# FROM FIVE DIMENSIONS TO MANY: LARGE LANGUAGE MODELS AS PRECISE AND INTERPRETABLE PSYCHOLOGICAL PROFILERS

**Yi-Fei Liu[1], Yi-Long Lu[2], Di He[3,5,*] & Hang Zhang[1,4,6,7,*]**

## ABSTRACT

Psychological constructs within individuals are widely believed to be interconnected. We investigated whether and how Large Language Models (LLMs) can model the correlational structure of human psychological traits from minimal quantitative inputs. We prompted various LLMs with Big Five Personality Scale responses from 816 human individuals to role-play their responses on nine other psychological scales. LLMs demonstrated remarkable accuracy in capturing human psychological structure, with the inter-scale correlation patterns from LLM-generated responses strongly aligning with those from human data ($R^2 > 0.88$). This zero-shot performance substantially exceeded predictions based on semantic similarity and approached the accuracy of machine learning algorithms trained directly on the dataset. Analysis of reasoning traces revealed that LLMs use a systematic two-stage process: First, they transform raw Big Five responses into natural language personality summaries through information selection and compression. Second, they generate target scale responses based on reasoning from these summaries. For information selection, LLMs identify the same key personality factors as trained algorithms, though they fail to differentiate item importance within factors. The resulting compressed summaries are not merely redundant representations but capture synergistic information—adding them to original scores enhances prediction alignment, suggesting they encode emergent, second-order patterns of trait interplay. Our findings demonstrate that LLMs can precisely predict individual participants' psychological traits from minimal data through a process of abstraction and reasoning, offering both a powerful tool for psychological simulation and valuable insights into their emergent reasoning capabilities.

## 1 INTRODUCTION

Understanding the human nomothetic network—the complex web of how psychological traits correlate (Cronbach & Meehl, 1955)—is a central ambition of psychology and a key frontier in computational social science (Karvelis et al., 2023; Ziems et al., 2024). Capturing this structure accurately thus serves as a profound benchmark for an AI's ability to reason about human nature. Large Language Models (LLMs) have emerged as promising candidates (Serapio-García et al., 2023), demonstrating powerful capabilities for group-level simulation in social and economic scenarios (Aher et al., 2023; Dillion et al., 2023). Concurrently, advancements in individual-level simulation have enabled models to simulate specific personality traits (Jiang et al., 2024) using techniques like in-context learning (Choi & Li, 2024). Furthermore, these capabilities have been integrated into large-scale agent frameworks (Pan et al., 2024), with landmark studies demonstrating LLMs' potential for mimicking real-world individuals with high fidelity (Park et al., 2024). Despite these successes in mimicking behaviors, the source of this capability remains a critical open question (Dong et al.,

---

[1]Peking-Tsinghua Center for Life Sciences, Peking University; [2]State Key Laboratory of General Artificial Intelligence, BIGAI, Beijing, China; [3]State Key Laboratory of General Artificial Intelligence, Peking University; [4]School of Psychological and Cognitive Sciences and Beijing Key Laboratory of Behavior and Mental Health, Peking University; [5]School of Intelligence Science and Technology, Peking University; [6]PKU-IDG/McGovern Institute for Brain Research, Peking University; [7]Key Laboratory of Machine Perception (Ministry of Education), Peking University, Beijing, China.
[*]Corresponding authors. Emails: *dihe@pku.edu.cn, hang.zhang@pku.edu.cn*.

2025): does it stem from genuine psychological reasoning, or from sophisticated pattern matching on inputs with strong semantic overlap (Messeri & Crockett, 2024)?

However, designing a convincing experiment to address this question is not easy. We believe it should meet the following key criteria. First, the task should, as far as possible, rule out the possibility that LLMs exploit semantic pattern-matching strategies (Messeri & Crockett, 2024; Suh et al., 2024) by using controlled, low-overlap inputs and requiring item-level predictions for individuals. Second, the data analysis should move beyond evaluating LLMs on isolated trait-to-trait predictions, and instead provide a comprehensive account of the correlation structure among psychological traits. Third, the test should be evaluated across different models (Samuel et al., 2024; Tjuatja et al., 2024), ensuring that the conclusion is widely applicable rather than model-specific artifacts. Finally, the test should enable a degree of process-level interpretability, moving beyond performance metrics to distinguish between genuine psychological reasoning and sophisticated pattern matching that could yield similar performance scores. Given that the faithfulness of Chain-of-Thought rationales remains contested (Wei et al., 2022; Turpin et al., 2023), this necessitates complementary approaches to reveal the underlying cognitive processes driving LLM predictions.

To fulfill these criteria, here we designed a new psychometric paradigm and data analysis pipeline. We tasked a diverse suite of LLMs with a challenging, zero-shot prediction task: given only the 20 item-level answers (on a 5-point response scale from strongly disagree to strongly agree) from an individual's Big Five personality inventory—a widely-used measure of five core personality dimensions—LLMs were required to predict the individual's answers on each item of nine other, distinct psychological scales (questionnaires). To evaluate LLM predictions, we compared them against human participants' actual responses on these psychological scales using a dataset where each participant completed all measures. Different from prevalent methodologies that focus on first-order prediction accuracy—assessing how well models directly replicate specific behavioral instances or trait scores (Jiang et al., 2024; Park et al., 2024; Zhu et al., 2025)—we argue that simply correlating model predictions with actual scores would be methodologically flawed in this context, as such correlations are constrained by the ground-truth relationships between the Big Five and target scales. Instead, we developed a second-order morphism measure that compares the inter-scale correlation patterns in LLM predictions versus human data, using the regression coefficient between these patterns as our alignment metric. To further achieve interpretability, we analyzed the reasoning traces of LLMs with reasoning capabilities using a meta-prompt approach, where annotation models parsed the reasoning outputs to identify which specific Big Five items influenced each prediction and the underlying information-processing stages.

Our experiments on LLMs yield two interconnected findings characterizing both LLM performance and the underlying process. The inter-scale correlations elicited from LLM predictions showed strong linear alignment with human data ($R^2 > 0.88$). This zero-shot performance substantially exceeded predictions based on semantic similarity and approached the accuracy of machine learning algorithms trained directly on the dataset, indicating that LLMs can accurately reconstruct the nomothetic network of how individuals' different psychological traits covary. Our analysis of the models' reasoning traces reveals this capability is driven by a two-stage process: First, LLMs transform raw Big Five responses into natural language personality summaries through information selection and compression; second, they generate target scale responses by reasoning from these summaries. For information selection, LLMs identify the same key personality factors as trained algorithms, though they fail to differentiate item importance within factors. The resulting compressed summaries prove sufficient for prediction, achieving similar alignment performance when used as input in place of the original Big Five responses, and even superior alignment when used as an addition.

Our work demonstrates that LLMs possess genuine psychological reasoning capabilities rather than mere semantic pattern matching, as evidenced by their ability to accurately reconstruct the correlational structure of psychological traits from minimal personality data through systematic abstraction processes. By developing novel evaluation methods that bypass semantic overlap and revealing the mechanistic underpinnings of LLM psychological reasoning, we provide both a powerful tool for psychological simulation and insights into the interpretability of emergent AI reasoning capabilities.

## 2 RELATED WORK

**LLMs as Human Simulators**  Many studies on LLMs involve personality tests, but for different purposes. Some measure psychological traits exhibited by the models themselves (Pellert et al., 2024; Sorokovikova et al., 2024; Tjuatja et al., 2024; Dong et al., 2025), while others use LLMs to role-play human participants, either to demonstrate their capability to replicate aggregate human behaviors (Aher et al., 2023; Argyle et al., 2023; Dillion et al., 2023; Horton, 2023; Santurkar et al., 2023) or to generate individualized agents with specified personality profiles (Choi & Li, 2024; Jiang et al., 2024; Petrov et al., 2024). Like our work, some of these studies use Big Five personality traits as input (Vu et al., 2024; Li et al., 2025; Wang et al., 2025b). Notably, Park et al.'s "Generative Agent Simulations of 1,000 People" (Park et al., 2024) demonstrates the possibility of simulating individual-level agentic behavior that aligns with humans across multiple dimensions of psychological traits and behavioral attitudes. Our work extends this line of work in three key aspects. First, whereas prior works primarily evaluate first-order prediction accuracy (i.e., how accurately the model mimics a specific trait), we perform a second-order analysis investigating whether LLMs can accurately reconstruct the entire psychological correlational network between traits. Second, our use of sparse inputs isolates the model's inferential capability from the memory retrieval common in data-rich, open-world settings (Park et al., 2024). Finally, we analyze reasoning traces to reveal the cognitive processes driving this capability, rather than merely evaluating outcomes.

**Deconstructing Reasoning Mechanisms in LLMs**  The advent of Chain-of-Thought (CoT) prompting (Kojima et al., 2022; Wang et al., 2022; Wei et al., 2022), particularly in reasoning-enhanced models (Bi et al., 2024), suggests that models could externalize their reasoning steps. However, a key debate surrounds the faithfulness of these rationales: whether they truly guide the model's reasoning or are merely plausible post-hoc justifications (Lanham et al., 2023; Turpin et al., 2023; Paul et al., 2024; Arcuschin et al., 2025)? Our methodology offers insights into this debate. By analyzing LLMs' self-generated reasoning trace, we find that LLMs spontaneously generate an intermediate representation based on high-level personality factors before predicting specific item responses. This suggests that LLMs' reasoning is not a passive retrieval process, as in Self-RAG (Asai et al., 2024), but an active process of information compression. LLMs synthesize verbose item data into a dense, low-dimensional summary, a representation our analysis confirms is instrumental in shaping the final prediction. The functional utility of this summary is further supported by providing it back to the model as additional input, which enhances prediction performance.

**Evidence for a Cognitive Hierarchy in LLMs**  A central question in AI research concerns the quality and depth of cognition in LLMs, with debates often polarized between claims of genuine understanding (Ichien et al., 2024) and the "stochastic parrot" hypothesis (Bender et al., 2021). A more productive framework, inspired by cognitive science, is to investigate the existence of a cognitive hierarchy (Landauer & Dumais, 1997; Huber & Niklaus, 2025; Wang et al., 2025a), which posits that cognition comprises tiered processes of increasing sophistication, from simple association to abstraction and rule-based generalization (Eppe et al., 2022; Graham & Granger, 2025). Addressing the methodological gap in how to dissect these strata, our work introduces a novel psychometric paradigm to move the conversation from whether LLMs reason to how and at what level of abstraction they operate. Our findings chart a path up this hierarchy, demonstrating that LLMs' cognitive process (1) transcends surface-level statistical association, (2) prioritizes abstract conceptual structure over specific item-level details, and (3) performs a novel form of theoretical idealization, actively refining noisy inputs into theory-consistent representations.

## 3 EXPERIMENT 1: LLMS' RECONSTRUCTION OF HUMAN PSYCHOLOGICAL STRUCTURES

The aim of Experiment 1 is to test whether LLMs can reconstruct the entire nomothetic network from sparse, quantitative personality inputs such as Big Five personality scores, distinguishing between genuine reasoning and pattern matching. A dataset comprising the psychological test results of 816 Chinese participants, collected online during the COVID-19 pandemic as part of a longitudinal diary study on epidemic perceptions (Lu et al., 2023), served as the ground truth. The dataset consists of each participant's responses to the Big Five personality scale (Topolewska et al., 2014),

which provided the input for the LLMs, alongside scores from nine other psychological scales: the Perceived Stress Scale (Cohen et al., 1983; Leung et al., 2010), the Simplified Coping Style Questionnaire (Xie, 1998), the State-Trait Anxiety Inventory (Spielberger et al., 1971), the Self-Compassion Scale (Neff, 2003), the Psychological Resilience Scale (CD-RISC) (Connor & Davidson, 2003), the Intolerance of Uncertainty Scale (Buhr & Dugas, 2002), the Emotion Regulation Questionnaire (Gross & John, 2003), the Risk Perception & Behavior Questionnaire, and the Future Time Perspective Scale (Carstensen & Lang, 1996). Collectively, these instruments assess a wide spectrum of constructs related to emotional well-being, adaptive functioning, and cognitive styles.

We evaluated a comprehensive suite of state-of-the-art LLMs, including their "Chat" and reasoning-enhanced "Thinking" variants where applicable: DeepSeek's DeepSeek-V3.1 (Liu et al., 2024), OpenAI's GPT-5 (OpenAI, 2025), Anthropic's Claude 3.7 Sonnet (Anthropic, 2025), Google's Gemini 2.5 Flash (Comanici et al., 2025), Zhipu AI's GLM-4.5 (Zeng et al., 2025), Moonshot AI's Kimi K2 (Team et al., 2025), and Alibaba's Qwen3-235B (Yang et al., 2025). We benchmarked their performance against traditional machine learning models (e.g., K-Nearest Neighbors, SVM, Linear Regression) and a Semantic Similarity model based on BAAI's bge-reranker-large model[1] (BAAI, 2024), a cross-encoder for relevance scoring (details in Appendix B.4).

## 3.1 TESTING PROCEDURES FOR LLMS

The core procedure of our experiment, as illustrated in Figure 1, consists of two phases: (1) **Prediction Generation (Per-Individual Task).** For each of the 816 individuals in our dataset, we tasked the LLM with a role-playing prediction. Each task involved providing the model with the individual's 20 item-level scores from the Big Five inventory. This served as the sole information source for the model to predict that same person's responses on all items across the nine other psychological scales. (2) **Structural Comparison Analysis (Dataset-Level Analysis).** After the model generated the complete dataset of predictions, we performed the following structural analysis. We computed the Pearson correlation matrix for every pair of psychological scale sub-factors (i.e., the individual dimensions that make up broader psychological measures) within the LLM-generated data. This matrix was then compared against a benchmark correlation matrix, which was calculated using the same method on the human ground-truth data, to evaluate the overall structural fidelity of the model's psychological inferences.

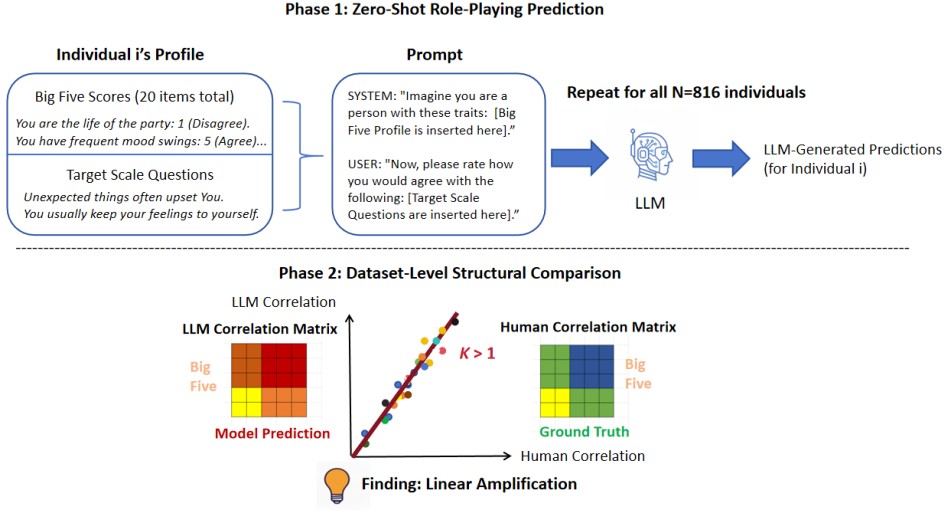

Figure 1: Experiment 1's procedure. Phase 1 (Per-Individual): The LLM role-played each participant based on the individual's Big Five scores to predict responses on nine other target scales. Phase 2 (Dataset Level): We computed correlation matrices between all psychological scale components for both LLM predictions and human ground truth data, and then compared the resulting correlation matrices, which reveals a structural amplification in LLMs' reconstructed psychological structures.

---

[1]The model was accessed via Alibaba Cloud's OpenSearch platform, where it is identified as "ops-bge-reranker-larger". We refer to it by its original BAAI name for clarity.

Our primary analysis focuses on comparing these correlation matrices to assess whether the model captures the underlying structure of human psychological trait relationships, rather than simply evaluating individual prediction accuracy. As noted in the Introduction, this approach provides a more robust test of whether the model has internalized the human psychological network, independent of the actual strength of relationships in the ground truth data.

## 3.2 LLMs Reconstruct a Linearly Amplified Psychological Structure

This structural comparison reveals that LLMs not only accurately replicate the correlational structure of human psychological traits, but also reconstruct an idealized, amplified version of it. First, when the correlation matrix from the model-generated scores is compared against that of human data, the model-generated correlations (Gemini 2.5) consistently match the sign of human data but are more saturated (further from zero) than their human counterparts (see Figure 2, top-left panel).

Second, to quantify this observation, we regressed the LLM's inter-scale correlations against the human data. As shown in Figure 2 (top-right), Gemini 2.5 demonstrates an exceptionally strong linear relationship ($R^2 = 0.92, p < .001$) with a regression slope ($k$) of 1.42, significantly greater than 1.0. This indicates that LLMs systematically overestimate the strength of correlations between psychological traits—a phenomenon we term **structural amplification**. This should be distinguished from "bias amplification" (Fernández et al., 2023; Taori & Hashimoto, 2023; Wang et al., 2024), which typically describes a first-order effect where models exaggerate specific, pre-existing societal biases from training data (e.g., gender stereotypes). In contrast, the structural amplification we identify is a second-order effect concerning the entire relational network of traits, rather than the strengthening of a specific biased association. To confirm the model constructs a coherent internal psychological network rather than just input-output mapping, we analyzed the correlational structure among predicted target scales themselves (excluding Big Five inputs). The amplification effect persists with remarkable strength ($k = 1.41, R^2 = 0.91, p < .001$), confirming the model builds a coherent and internally amplified representation of the entire psychological structure.

Third, we verified this is a general property of LLMs. All tested LLMs (Figure 2, bottom-left) exhibit an amplification coefficient $k > 1.0$ (see Appendix B.2 for their scatter plots). Crucially, all LLMs' amplification coefficients surpassed that of baseline models like a k-Nearest Neighbors (KNN) model and, notably, the Semantic Similarity model. This suggests the models' performance stems from a process more sophisticated than simple retrieval or surface-level semantics. To investigate the functional implications of this phenomenon, we correlated each model's amplification coefficient ($k$) with its predictive performance, a metric derived from the mean Pearson correlation between its predictions and the ground-truth scores across factors. The scatter plot in the bottom-right panel reveals a near-perfect positive linear relationship between these two variables ($R^2 = 0.95, p < .001$). The greater a model amplifies the underlying structure, the better it predicts human traits, suggesting this amplification is not an artifact but a core functional mechanism.

## 3.3 Validating structural amplification: Statistical Significance and Robustness

We performed two further analyses to validate the observed linear structural amplification is genuine and robust. First, we established statistical significance using a 1,000-trial permutation test. By shuffling the model-generated correlation vectors, we created an empirical null distribution for our key statistics ($R^2$ and Kendall's $\tau$). The originally observed statistics were extreme outliers relative to this null distribution ($p < .001$), unlikely to arise from pure chance.

Second, to ensure the structural amplification arises from genuine psychological reasoning and is not merely an artifact of the prompt's specific phrasing or structure, we examined the model's sensitivity to task framing and input organization (Oren et al., 2023). We tested three conditions: (1) our original "Standard Order" setup, where the Big Five items were presented in a fixed sequence; (2) a "Random Order" condition, where the sequence of the 20 Big Five items was shuffled for each trial to disrupt potential order effects and (3) a "Single Question" condition, where each target item was predicted individually to rule out artifacts from batch-prompting. The amplification coefficient for Gemini 2.5 remained exceptionally stable across these conditions ($k = 1.42, 1.41,$ and $1.42,$ respectively), demonstrating that it is a robust feature of the model's reasoning process. Detailed results and figures for these validation tests are provided in Appendix C.1.

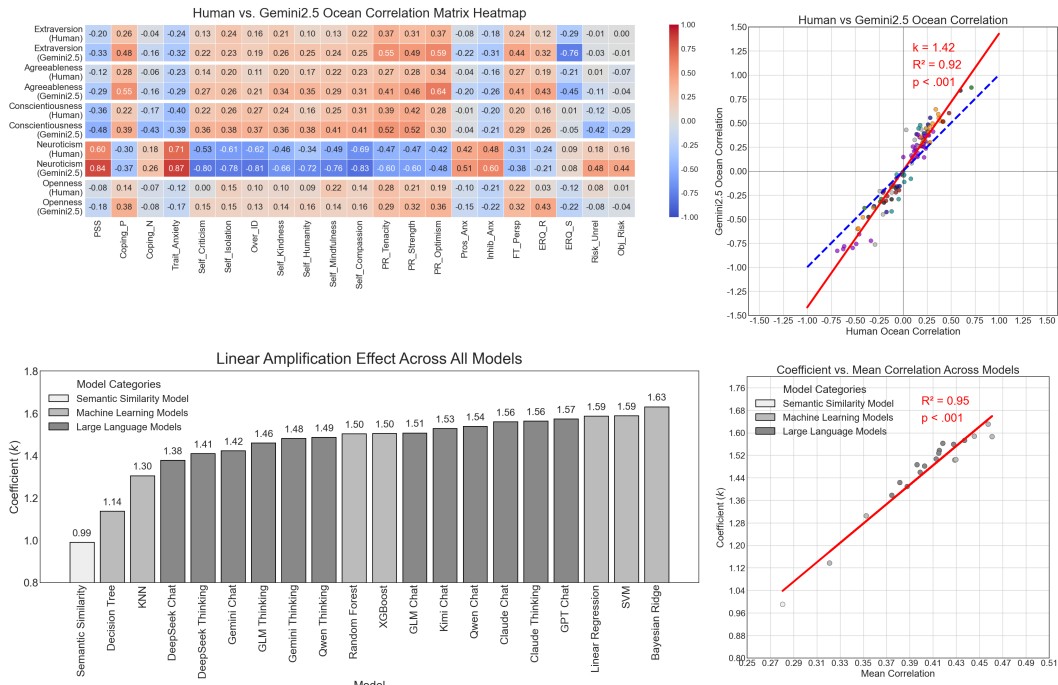

Figure 2: LLMs' structural amplification of the correlational structure of human psychological traits. Top-left panel: a heatmap comparing the correlations from human data with those predicted by Gemini 2.5 (paired rows), separately for each of the Big Five personality factors (rows) and the target psychological scales or their sub-scales (columns). Top-right panel: Gemini 2.5's correlations against human data correlations, revealing a strong linear relationship ($R^2 = 0.92$) with an amplification slope ($k = 1.42$). Each dot denotes a pair of correlations in top-left panel (color codes different target scales, see Appendix D for a complete list). Bottom-left panel: All tested LLMs exhibit an amplification coefficient $k > 1.0$, consistently outperforming retrieval (KNN) and semantic similarity models. Bottom-right panel: a near-perfect linear relationship ($R^2 = 0.95$) between a model's amplification coefficient ($k$) and its predictive performance. Each dot denotes one model.

## 3.4 Explaining Structural Amplification: The Idealization Hypothesis

To understand the source of the structural amplification, we tested the hypothesis that it stems from LLMs acting as "idealized participants" by filtering the random measurement error inherent in human self-reports. According to classical test theory, such random noise attenuates the observed correlations between psychological scales (Spearman, 1961; Nunnally, 1975). Our investigation into this hypothesis proceeds in two stages: first laying the theoretical foundation with reliability analysis, and then providing empirical validation through convergent experiments.

Our analysis begins with the theoretical support for the idealization hypothesis, grounded in reliability metrics. As detailed in Appendix C.4.1, we first compared the internal consistency (Cronbach's Alpha) of both human- and LLM-generated data. The LLM data exhibited significantly higher reliability (mean $\alpha_{\text{LLM}} = 0.87$) than the human data ($\alpha_{\text{Human}} = 0.75$), suggesting that LLMs produce less noisy responses. Furthermore, we found that LLM reliability profiles showed strong inter-model convergence (mean inter-LLM MSE = 0.0060 vs. mean LLM-to-Human MSE = 0.0357). This suggests that different LLMs converge on a common, idealized response model that differs from human patterns, positioning structural amplification as a direct consequence of higher data fidelity.

We then sought empirical validation for this noise-filtering hypothesis through two convergent experimental analyses. From the human side, we isolated a more "attentive" subgroup ($N = 309$) by filtering out participants with fast response times. This less-noisy human data produced an inherently stronger correlation structure that was significantly closer to the LLM's output ($k = 1.08$ when compared to the full sample). Conversely, from the model side, we established a near-causal

link through intervention. By systematically injecting increasing levels of Gaussian noise into a baseline model's predictions, we observed a clear dose-response relationship: as noise increased, the structural amplification effect was progressively attenuated ($k$ decreased from 1.55 to 1.12). This convergence of evidence—where removing noise from human data and adding it to model predictions produce opposite, predictable effects—provides robust empirical support for interpreting structural amplification as a process of idealized abstraction (see Appendix C.4.2 for details).

## 4 EXPERIMENT 2: DECONSTRUCTING THE REASONING MECHANISM

Experiment 1 revealed that LLMs exhibit a systematic "structural amplification" when simulating personality networks. However, this finding describes a macroscopic outcome, leaving the internal process a black box. Experiment 2 is designed to open this black box. By analyzing the models' reasoning traces, we aim to uncover the cognitive processes driving this phenomenon, thereby enhancing the explainability of their behavior. Specifically, we move from qualitative observation to a quantitative, testable framework that addresses two key questions: (1) How do models perform information selection from sparse inputs, and does this process align with psychological-conceptual structures? (2) Is the natural language summary generated during reasoning merely a byproduct, or is it a predictively potent form of information compression?

### 4.1 INFORMATION SELECTION: A CONCEPT-DRIVEN STRATEGY

Our analyses reveals that LLMs use what we may call a "concept-driven" reasoning strategy, prioritizing forming high-level conceptual understandings (e.g., personality factors) from the raw data to guide their predictions, rather than operating on isolated input items directly.

**Methodology** To parse the complex reasoning traces collected from Experiment 1, we used the methodology illustrated in Figure 3. For each specific reasoning trace accompanying a prediction, the LLM that generated it is referred to as "Reasoning Model" (e.g., Claude3.7-Thinking, GLM4.5-Thinking). To deconstruct the Reasoning Model's decision-making process while controlling for the interpretive biases of any single annotator, we parsed each trace using a separate, diverse suite of "Annotation Models", which included DeepSeek-V3.1, Qwen3-235B, GLM-4.5, and Claude 3.7 Sonnet. Each Annotation Model generated a 20-dimensional attribution distribution that quantified the perceived importance of each Big Five input item. These individual distributions were then averaged to produce a single, robust, and de-biased attribution vector for each reasoning trace. Finally, we benchmarked these averaged distributions against the feature importance weights derived from an outperforming Bayesian Ridge Regression model trained on ground truth.

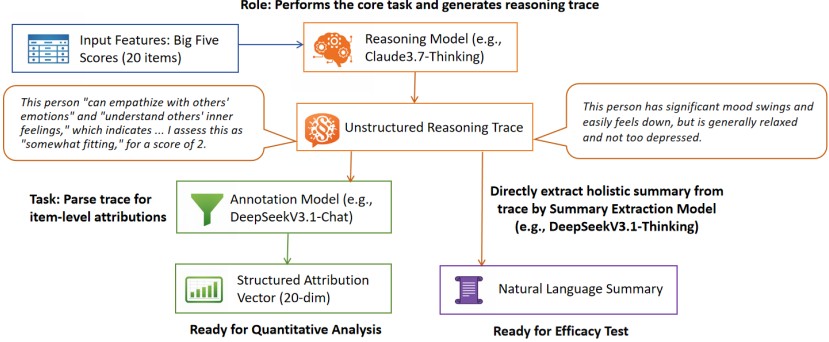

Figure 3: Flowchart of the "Reasoning-to-Annotation" analyses in Experiment 2. Each reasoning trace comes from Experiment 1, where "Reasoning Models" use the 20 input scores to generate predictions along with reasoning traces. These traces are processed in two parallel analyses: an "Annotation Model" parses each trace to create a structured attribution vector (addressing our first research question), while the summary within that same trace is used to predict the outcome, testing its predictive potency (addressing our second research question).

**Convergent Attribution Strategy** Our analysis reveals a remarkable cross-model consensus in attribution strategies. We performed a pairwise analysis across all combinations of Reasoning and Annotation models, and the resulting attribution distributions showed remarkable cross-model consensus. As detailed in the Appendix C.2.1, heatmaps of cross-model Pearson correlation coefficients and Kullback-Leibler (KL) divergences confirm this alignment. With self-comparisons excluded, the average Pearson correlation between the attribution vectors of different Reasoning Models was 0.934, and the average KL divergence was exceptionally low at 0.0475. This strong consensus indicates that different reasoning models independently reach the same strategy for mapping psychological traits, and this finding is not an artifact of the annotation process.

**Factor-Level Accuracy Despite Item-Level Confusion** To dissect this convergent strategy, we benchmarked the models' averaged attribution profiles against two distinct baselines at two levels of granularity (Figure 4). These baselines were derived by fitting a Bayesian Ridge model to two different targets: the Human Ground Truth baseline was fitted to actual human responses, while the Semantic Similarity baseline was fitted to predictions from our semantic similarity model (detailed in Appendix B.4). This three-way comparison allows us to test whether LLM reasoning aligns more with human psychological patterns or with semantic associations.

At the fine-grained item level, the analysis reveals a diffuse and inconsistent picture. The average Pearson correlation between the LLM attributions and the Human Ground Truth baseline is poor ($r = 0.207$). Similarly, the correlations between LLM attributions and the Semantic Similarity baseline ($r = 0.087$), and between the Human and Semantic baselines themselves ($r = 0.201$), are also weak. This indicates that at the level of individual items, no clear, consistent attribution strategy emerges across any of the models or baselines.

The distinction becomes stark at the coarse-grained factor level. Here, the LLM attribution profiles align almost perfectly with the Human Ground Truth baseline, achieving an average Pearson correlation of $r = 0.981$. In sharp contrast, the correlation between LLM attributions and the Semantic Similarity baseline is substantially lower ($r = 0.790$), a value nearly identical to the correlation between the Human and Semantic baselines ($r = 0.787$).

This clear dissociation uncovers a key insight into the models' reasoning: LLMs robustly identify the correct high-level personality factor (e.g., Neuroticism) in a way that mirrors human-like conceptual importance, significantly diverging from a strategy based on mere semantic similarity. While they struggle to differentiate the importance of specific items within that factor, their high-level abstraction process supports the hypothesis of a top-down, concept-driven inference, rather than one guided by surface-level word associations.

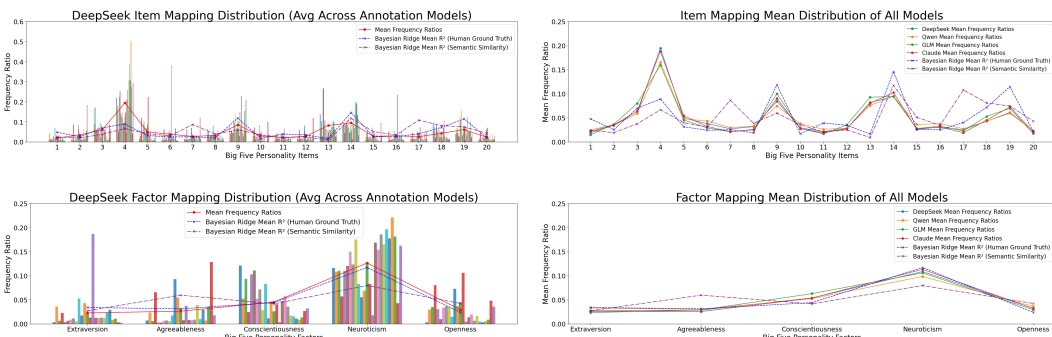

Figure 4: A concept-driven information selection strategy. The plots compare the models' averaged attributions (solid lines) against a Human Ground Truth (dashed blue) and a Semantic Similarity (dashed purple) baseline. The top row shows item-level attributions, where each bar represents a subscale, revealing weak and inconsistent alignment. The bottom row shows factor-level attributions, revealing that LLM attributions align almost perfectly with the Human Ground Truth, but starkly diverge from the Semantic Similarity baseline. This evidence supports that models identify high-level factors (e.g., Neuroticism) based on a human-like conceptual understanding, rather than on surface-level semantics. Each colored bar denotes one subscale, with color codes following that of the scatter plot in the top-right panel of Figure 2.

## 4.2 INFORMATION COMPRESSION: THE POWER OF ABSTRACT SUMMARIES

**Methodology**   Beyond selecting items, reasoning-enabled models also generate holistic, natural-language summaries of the personality profile. To quantify the predictive power of these abstract summaries (e.g., concluding that the provided scores reflect the risk perception of *"an individual who is sensitive, prone to worry, imaginative, and relatively pessimistic, often overestimating the probability of negative events"*), we designed an experiment with three information conditions:

1. ScoreOnly: The baseline condition from Experiment 1, where the model receives only the 20 Big Five numerical scores.

2. SummaryOnly: The model receives only the generated natural-language summary.

3. Summary+Score: The model receives both the Big Five scores and the summary, and is instructed to use all available information.

**Summaries as a Potent and Sufficient Information Vehicle**   This analysis was aimed to understand the role of the natural-language summary—the brief, narrative description the model synthesizes to guide its final predictions. The central finding, as seen in Figure 5, is the remarkable efficacy of this summary alone. When comparing the SummaryOnly condition to the ScoreOnly condition, we found that the structural amplification effect remained remarkably robust. This demonstrates that the natural-language summary is not just a useful aid, but a highly potent and sufficient compression of the original 20-item numerical input. The model is able to reconstruct the vast majority of the personality structure from this compressed linguistic representation alone.

Furthermore, we observed an unexpected synergistic effect. The Summary+Score condition consistently yielded the highest amplification multiplier for every model (right panel of Figure 5). The fact that performance improves when adding a summary derived from the scores themselves indicates that the summary is not merely a redundant compression. Instead, it appears to contain emergent, second-order information—a conceptual gestalt—synthesized during the model's reasoning process. Crucially, this enhancement in the amplification coefficient ($k$) is functionally significant, as it corresponds with an increase in predictive performance. For all models, the mean predictive performance consistently followed the order: Summary+Score > ScoreOnly > SummaryOnly. An extended analysis across all 15 conditions (5 models $\times$ 3 information types) revealed a strong positive correlation between $k$ and predictive performance ($R^2 = 0.93, p < .001$), confirming that greater structural amplification systematically predicts better outcomes (see Appendix C.3 for details).

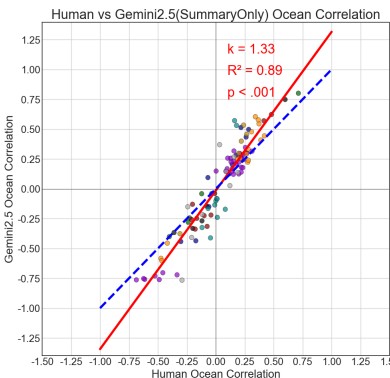
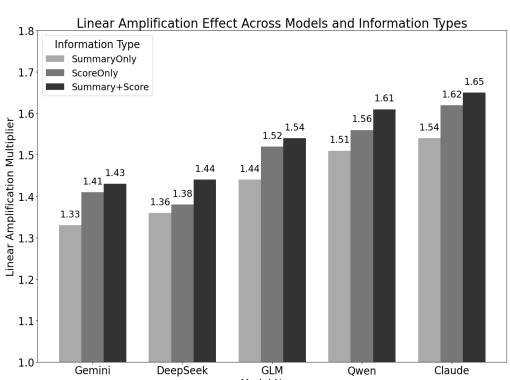

Figure 5: Analysis of the efficacy and synergy of LLM-generated summaries. The left panel shows the amplification effect persists even when using only the abstract summary ($R^2 = 0.91$). The right panel compares the amplification multiplier for all models across the three information conditions, demonstrating the synergistic value of adding the summary.

## 5 DISCUSSION

We investigated the capacity of Large Language Models to reason about psychological trait correlations from sparse, quantitative data. Our finding of good alignment between LLM predictions and

human psychological structure seemingly contrasts with Zhu et al. (2025), who found poor alignment when inferring personality from qualitative interviews. However, Zhu et al. assessed first-order prediction of specific traits, whose performance depends on ground truth correlation strength. We instead perform second-order structural analysis, comparing inter-scale correlation patterns between LLM predictions and human data. This approach reveals how LLMs preserve the entire correlational structure of psychological traits—capabilities that first-order prediction accuracy alone would miss.

Why do LLMs systematically "purify" the correlational structure of psychological traits? We conjecture that this stems from the model acting as an idealized participant, capable of bypassing the statistical noise inherent in human self-reports. Human responses are known to be contaminated by at least two distinct sources of noise: systematic biases from idiosyncratic response styles (Grimmond et al., 2025), and random measurement error arising from factors like inattention, fluctuating emotional states, or momentary misinterpretation of items (Nunnally, 1975). As detailed in Section 3.4, our investigation supports this noise-filtering hypothesis with convergent evidence. The higher internal consistency of LLM-generated data provides theoretical support, further substantiated by two empirical analyses: filtering noise from human data and injecting noise into model predictions (see Appendix C.4.2 for details). This evidence supports interpreting structural amplification as idealized abstraction driven by the model's ability to filter random statistical noise.

Having established that LLMs perform this abstraction, our analysis of reasoning traces reveals a two-stage process. First, during information selection, models prioritize high-level psychological factors over specific item details, a strategy our findings distinguish from reliance on surface-level semantics. This prioritization of abstract concepts over concrete data is consistent with both a cognitive Concept-Driven Strategy (Jackson, 2015) and the computational Information Bottleneck principle (Tishby et al., 2000), as the model generates a compressed representation by discarding less critical, item-level information. Meanwhile, information compression constructs a predictively potent natural-language summary that contains emergent, second-order information. Together, this integrated process of concept selection and information compression provides a clear account of how LLMs move beyond mere data replication to operate at a higher, more abstract level of reasoning.

Our work has several limitations that should be addressed in future research. First, the human dataset that our findings are based on is from a single cultural context that may lack broader demographic representativeness. Future work may apply our framework to diverse datasets to explore how cultural biases in training data might affect the idealized structure (Jakesch et al., 2023). Second, while we establish structural amplification as a general property of modern LLMs, we do not deconstruct the variance between them. Future work could investigate why certain models or architectures exhibit a stronger amplification effect, potentially linking this capacity to factors like model scale or fine-tuning methods. Third, the rich semantics of the synergistic summaries warrant further investigation. Finally, as our analyses are primarily observational, future studies should move towards interventional approaches,such as actively modulating the amplification effect to establish definitive causality.

## 6 CONCLUSION

This paper investigates the capacity of Large Language Models to reason about human individuals' psychological traits from sparse data. We identify a robust and counter-intuitive phenomenon we term structural amplification, where LLMs do not merely replicate but systematically idealize the correlational structure of human personality. We provide empirical evidence that this effect represents idealized abstraction, driven by the model's ability to filter statistical noise inherent in human self-reports. Our mechanistic analysis reveals that this abstraction involves concept-driven information selection and synergistic information compression that synthesizes predictively potent linguistic summaries. This work provides a mechanistic account of how LLMs transcend passive data replication and engage in active, abstract model construction.

## ETHICS STATEMENT

This research was conducted with strict adherence to ethical principles, prioritizing the privacy of human participants. Our study used a fully anonymized dataset from a previous study, which had been approved by the Institutional Review Boards of School of Psychological and Cognitive Sci-

ences at Peking University and Faculty of Psychology at Beijing Normal University. In that original data collection, all participants provided informed consent online. No personally identifiable information was accessed or used at any stage of our research, ensuring the confidentiality of all 816 participants.

## REPRODUCIBILITY STATEMENT

Core experimental details are provided in the Appendix. Specifically:

- **Data and Materials:** To facilitate reproducibility, the complete dataset, including anonymized participant responses, LLM-generated outputs, and the psychological scales used in our experiments, is available at our project repository.[2]
- **Prompts:** All prompt templates used for the main tasks (Experiment 1 & 2) and robustness checks are detailed in Appendix A and Appendix C.1.
- **Models:** A comprehensive list of all Large Language Models used in our experiments, including their specific model identifiers, is provided in Appendix B.1.
- **Baselines:** The implementation details for the semantic similarity baseline are described in Appendix B.4.

## ACKNOWLEDGMENTS

HZ was partly supported by National Natural Science Foundation of China (32471152), Clinical Medicine Plus X—Young Scholars Project of Peking University, the Fundamental Research Funds for the Central Universities (PKU2023LCXQ023 and PKU2024LCXQ046), and funding from Peking-Tsinghua Center for Life Sciences. DH was supported by National Science Foundation of China (NSFC62376007), National Science Foundation of China (under Key Project No. 92570203), Beijing Natural Science Foundation (Z250001) and Beijing Major Science and Technology Project under Contract no. Z251100008425004.

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

# APPENDIX

## A    EXPERIMENTAL PROMPTS

This section provides the English translations of the core prompt templates used in our experiments. Placeholders like '[Personality Profile]' and '[Target Scale Items]' are used to denote the parts of the prompt that were dynamically populated for each participant and task.

### A.1    MAIN TASK PROMPTS (EXPERIMENT 1 & 2)

### A.1.1    SCOREONLY CONDITION PROMPT

This is the standard prompt used to elicit predictions based only on the Big Five personality scores. The model is given a personality profile and asked to role-play that individual to answer questions from a target psychological scale.

```
[
  {
    "role": "system",
    "content": "Please imagine you are role-playing a specific person.
        ↪ The following are some descriptions of your personality. Each
        ↪ description has one of five levels of applicability: 'Strongly
        ↪ Disagree', 'Disagree', 'Neutral', 'Agree', 'Strongly Agree'.

    [20 Big Five personality items are inserted here, for example:]
    You are the life of the party: Disagree
    You sympathize with others' feelings: Agree
    You get chores done right away: Neutral
    You have frequent mood swings: Agree
    You have a vivid imagination: Agree
    ...

    Please rate the extent to which the person you are role-playing
        ↪ would agree with the following descriptions. If you 'Strongly
        ↪ Disagree', output 1. If 'Disagree', output 2. If 'Somewhat
        ↪ Disagree', output 3. If 'Neither Agree nor Disagree', output
        ↪ 4. If 'Somewhat Agree', output 5. If 'Agree', output 6. If
        ↪ 'Strongly Agree', output 7.

    Please use the format 'Description 1:(integer from 1 to
        ↪ 7);Description 2:(integer from 1 to 7)...' for your output."
  },
  {
    "role": "user",
    "content": "[Target scale items are inserted here, for example:]
    Description 1: When I want to feel more positive emotion (such as
        ↪ joy or amusement), I change what I'm thinking about.
    Description 2: I keep my emotions to myself.
    Description 3: When I want to feel less negative emotion (such as
        ↪ sadness or anger), I change what I'm thinking about.
    ..."
  }
]
```

### A.1.2    SUMMARYADDED CONDITION PROMPT

This prompt is identical to the `ScoreOnly` condition, but includes the model-generated summary as additional context. The model is explicitly instructed to use all available information.

```
[
  {
    "role": "system",
    "content": "Please imagine you are role-playing a specific person.
```

```
    The following are some descriptions of your personality...

    [The same 20 Big Five personality items as in ScoreOnly]

    The following is a supplementary summary generated by a model, which
        ↪ you may refer to as you see fit:
    [The LLM-generated abstract summary is inserted here, for example:]
    Key Points Summary:
    This person is likely pessimistic and prone to worry, being
        ↪ emotionally volatile and easily disheartened... these
        ↪ estimations reflect the risk perception of an individual who
        ↪ is sensitive, prone to worry, imaginative, and relatively
        ↪ pessimistic, often overestimating the probability of negative
        ↪ events
    ...

    Please rate the extent to which the person you are role-playing
        ↪ would agree with the following descriptions...
    [The rest of the prompt is identical to the ScoreOnly condition]"
  },
  {
    "role": "user",
    "content": "[Target scale items are inserted here]"
  }
]
```

### A.1.3 SUMMARYONLY CONDITION PROMPT

In this condition, the model receives only the abstract summary, without the original numerical scores.

```
[
  {
    "role": "system",
    "content": "Please imagine you are role-playing a specific person.
        ↪ The following is a summary of your personality:

    [The LLM-generated abstract summary is inserted here]

    Please rate the extent to which the person you are role-playing
        ↪ would agree with the following descriptions...
    [The rest of the output instructions are identical to the ScoreOnly
        ↪ condition]"
  },
  {
    "role": "user",
    "content": "[Target scale items are inserted here]"
  }
]
```

## A.2 ROBUSTNESS CHECK PROMPTS

### A.2.1 RANDOM ORDER CONDITION PROMPT

The prompt for this condition was identical to the ScoreOnly prompt, with the sole exception that the 20 Big Five personality items presented in the system message were randomly shuffled for each trial. The user message containing the target scale items remained unchanged.

### A.2.2 SINGLE QUESTION CONDITION PROMPT

This prompt was modified to test for artifacts from asking multiple questions at once. The model was prompted for each target item individually.

```
[
  {
    "role": "system",
    "content": "Please imagine you are role-playing a specific person...

    [The same 20 Big Five personality items as in ScoreOnly]

    Please rate the extent to which the person you are role-playing
        ↪ would agree with the following description... Output only a
        ↪ single integer from 1 to 7, with no other text."
  },
  {
    "role": "user",
    "content": "Description: [A single target scale item is inserted
        ↪ here]"
  }
]
```

## A.3 REASONING MECHANISM ANALYSIS PROMPTS (EXPERIMENT 2)

### A.3.1 ATTRIBUTION MAPPING PROMPT (FOR INFORMATION SELECTION ANALYSIS)

This meta-prompt was given to an "Annotation Model" to parse the reasoning trace of a "Reasoning Model" and identify which input items were used for a prediction.

```
[
  {
    "role": "user",
    "content": "We have provided the Big Five personality scores of a
        ↪ participant to a large language model and asked it to predict
        ↪ the participant's scores on the ERQ questionnaire items. Based
        ↪ on the model's reasoning content below, please identify which
        ↪ Big Five items the model relied on when giving its score for
        ↪ each ERQ description.

    Strictly follow the format 'Description 1: Item 2, Item 3;
        ↪ Description 2: Item 7, Item 15, Item 20...'. Do not output any
        ↪ other content.

    The following are the Big Five scores input to the model:
    [20 Big Five personality items with their scores]

    The following is the model's reasoning content:
    [The full reasoning trace from the 'Thinking Model' is inserted
        ↪ here]"
  }
]
```

### A.3.2 SUMMARY EXTRACTION PROMPT (FOR INFORMATION COMPRESSION ANALYSIS)

This prompt was used to have a model read its own reasoning trace and extract only the parts that constitute a holistic summary of the persona.

```
[
  {
    "role": "user",
    "content": "We have provided the Big Five personality scores of a
        ↪ participant to a model and asked it to predict scores on the
        ↪ relevant questionnaire. The following is the model's reasoning
        ↪ content.

    Please determine if the reasoning content includes a summary
        ↪ description of the person (pay attention to summary words like
        ↪ 'in summary', 'overall', 'synthesizing'). If it does, please
```

```
      ↪ output these summary descriptions exactly as they appear,
      ↪ without omitting any. If there are multiple summaries, include
      ↪ them all. Do not output any other content. If no summary is
      ↪ found, output 'None'.

   Reasoning Content:
   [The full reasoning trace from the 'Reasoning Model' is inserted
      ↪ here]"
  }
]
```

# B MODELS AND PERFORMANCE

## B.1 LIST OF LANGUAGE MODELS

For clarity and reproducibility, this section lists the large language models used. All models were accessed via API calls to the OpenRouter platform, and their identifiers in Table 1 follow its specific conventions.

Our study utilized two model conditions: a standard "Chat" mode and a "Thinking/CoT" (Chain-of-Thought) mode. Experiment 1 benchmarked all listed models for predictive performance, with most models being evaluated under both conditions. Experiment 2 then focused exclusively on analyzing the reasoning traces from the "Thinking/CoT" modes. This advanced mode was enabled either by calling a dedicated model identifier (e.g., with a :thinking suffix) or via an API parameter, as specified in the table.

Table 1: Large Language Models Evaluated in the Study.

| Developer | Short Name | Condition/Notes | Model Identifier |
|---|---|---|---|
| DeepSeek AI | DeepSeek V3.1 | Standard Chat | deepseek/deepseek-chat-v3.1 |
| DeepSeek AI | DeepSeek V3.1 | Thinking/CoT | deepseek/deepseek-chat-v3.1 |
| OpenAI | GPT-5 | Standard Chat | openai/gpt-5-chat |
| Anthropic | Claude 3.7 Sonnet | Standard Chat | anthropic/claude-3.7-sonnet |
| Anthropic | Claude 3.7 Sonnet | Thinking/CoT | anthropic/claude-3.7-sonnet:thinking |
| Google | Gemini 2.5 Flash | Standrd Chat | google/gemini-2.5-flash |
| Google | Gemini 2.5 Flash | Thinking/CoT | google/gemini-2.5-flash |
| Zhipu AI | GLM-4.5 | Standard Chat | z-ai/glm-4.5 |
| Zhipu AI | GLM-4.5 | Thinking/CoT | z-ai/glm-4.5 |
| Moonshot AI | Kimi K2 | Standard Chat | moonshotai/kimi-k2-0905 |
| Alibaba | Qwen3-235B | Standard Chat | qwen/qwen3-235b-a22b |
| Alibaba | Qwen3-235B | Thinking/CoT | qwen/qwen3-235b-a22b-thinking-2507 |

## B.2 INDIVIDUAL MODEL SCATTER PLOTS

To provide a comprehensive visualization of the structural amplification phenomenon across all evaluated large language models, this section presents scatter plots for each of the 12 large language models tested in our study. Each plot displays the linear relationship between the inter-scale correlations derived from model predictions and those from human ground-truth data, analogous to the Gemini 2.5 example shown in Figure 2 (top-right panel) of the main text.

The consistent pattern observed across all models reinforces our central finding: LLMs systematically reconstruct an amplified version of the human psychological correlational structure, with regression slopes ($k$) consistently exceeding 1.0. This cross-model consistency provides robust evidence that structural amplification is a general property of modern large language models rather than a model-specific artifact.

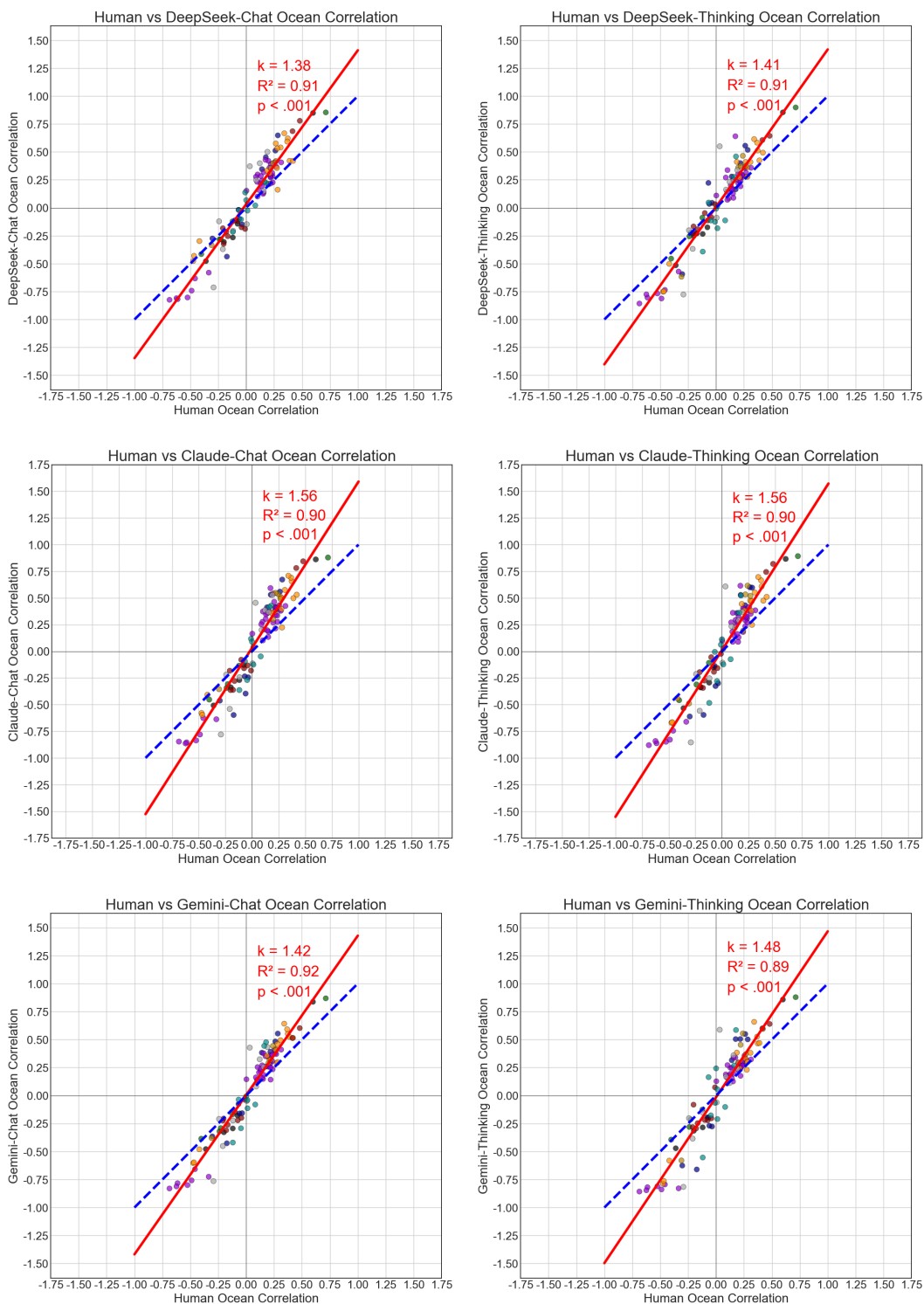

Figure 6: Scatter plots demonstrating the structural amplification effect for the first six evaluated large language models. Each plot shows the linear relationship between model-predicted inter-scale correlations and human ground-truth correlations, with regression slope $k > 1.0$ indicating systematic amplification of the psychological structure.

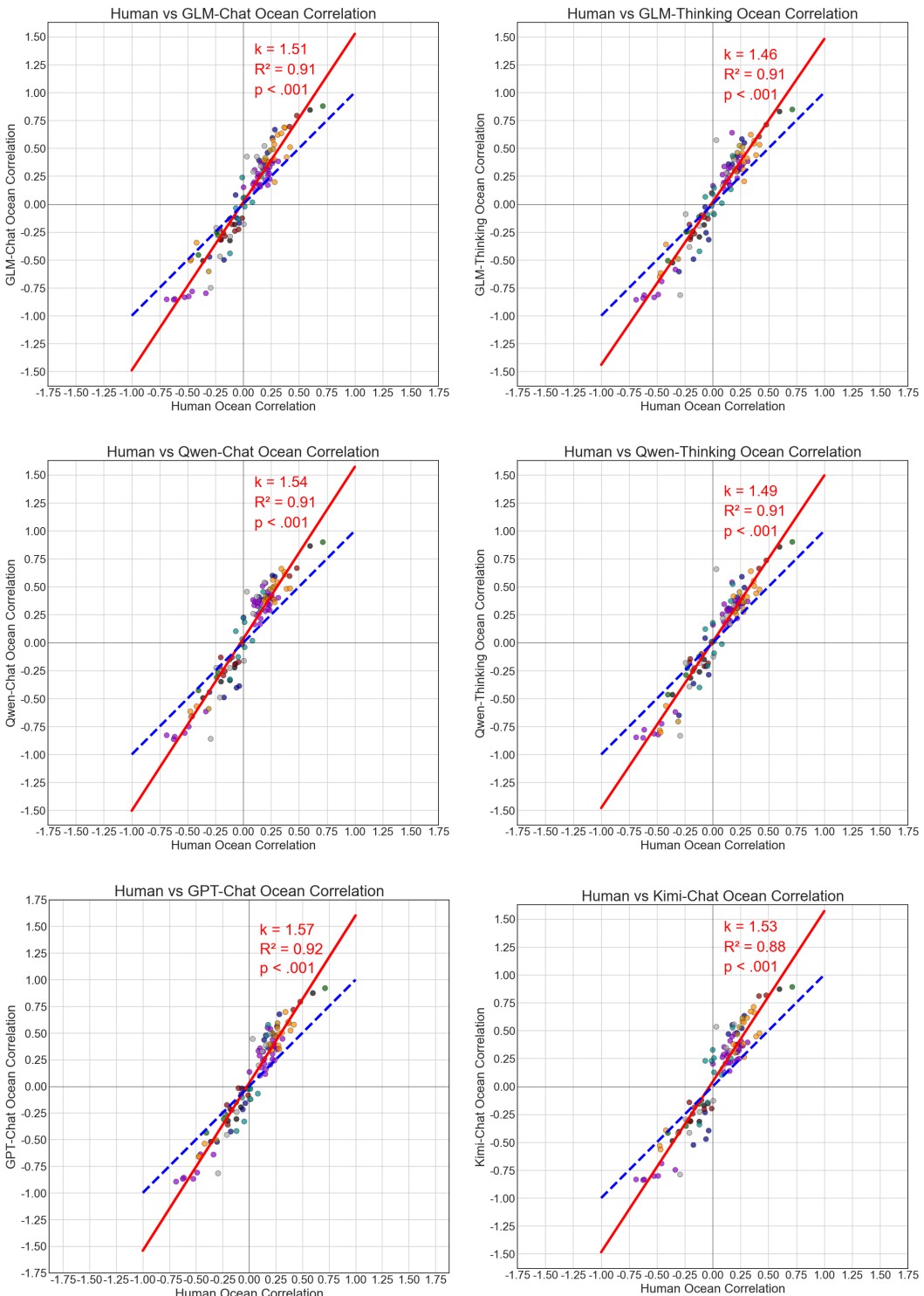

Figure 7: Scatter plots demonstrating the structural amplification effect for the remaining six evaluated large language models. Each plot shows the linear relationship between model-predicted inter-scale correlations and human ground-truth correlations, with regression slope $k > 1.0$ indicating systematic amplification of the psychological structure. The consistent pattern across all 12 models provides robust evidence for structural amplification as a general property of modern LLMs.

### B.3 DETAILED PREDICTIVE PERFORMANCE METRICS (HEATMAP)

To supplement the analysis of the structural amplification effect, this section provides a detailed breakdown of each model's predictive performance. Figure 8 visualizes the Pearson correlation coefficient ($r$) between each model's generated scores and the human ground-truth scores for every target psychological subscale. This heatmap offers a granular view of model capabilities, revealing which models excel at predicting specific psychological constructs and providing the detailed evidence that supports the aggregate performance rankings discussed in the main text.

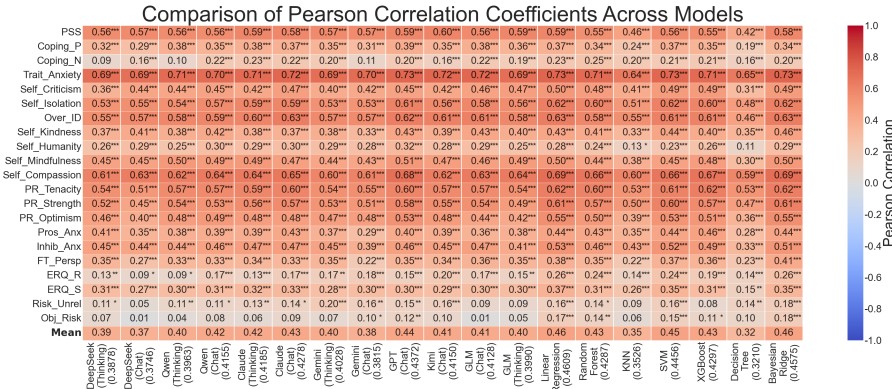

Figure 8: Detailed heatmap of predictive performance across all models and target psychological scales. The metric shown is the Pearson correlation coefficient ($r$).

### B.4 SEMANTIC SIMILARITY BASELINE

To test if structural amplification stems from surface-level semantics, we built a non-reasoning baseline. We used a re-ranking model, specifically the `ops-bge-reranker-larger`, which is a BGE-based cross-encoder model served through Alibaba Cloud's OpenSearch platform for document relevance scoring. This model generated semantic similarity scores that served as fixed weights in a linear model to predict target scale scores from a participant's Big Five input. Crucially, predictions for reverse-scored items were inverted to ensure directional correctness.

As shown in Figure 9, this baseline yielded an amplification coefficient of only $k = 0.99$ with a weak fit ($R^2 = 0.52$). A coefficient near 1.0 indicates a failure to amplify, merely replicating the input data's structure. This finding provides strong evidence that the LLM's amplification phenomenon is not a byproduct of simple semantic matching but stems from a more sophisticated, abstract reasoning process.

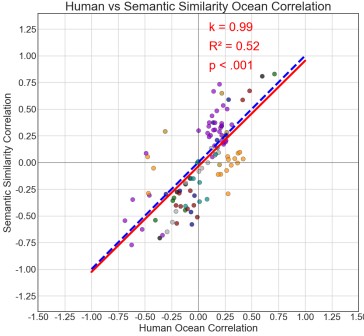

Figure 9: Analysis of the Semantic Similarity baseline. The regression slope ($k = 0.99$) is nearly 1.0, indicating a failure to produce the structural amplification effect. The poor linear fit ($R^2 = 0.52$) further confirms that LLM performance transcends surface-level semantics.

## C SUPPLEMENTARY ANALYSES FOR MAIN FINDINGS

### C.1 VALIDATING STRUCTURAL AMPLIFICATION: SIGNIFICANCE AND ROBUSTNESS

To ensure the observed structural amplification is a genuine and robust feature of the models' reasoning, we conducted two further validation analyses.

**Statistical Significance.** We conducted a 1,000-trial permutation test, shuffling the LLM-generated correlation vectors to construct an empirical null distribution for our key statistics ($R^2$ and Kendall's $\tau$). As shown in Figure 10, the originally observed statistics were extreme outliers, falling far outside the range of values expected under the null hypothesis ($p < .001$). This confirms the structural amplification effect is a highly significant phenomenon and not a product of random chance.

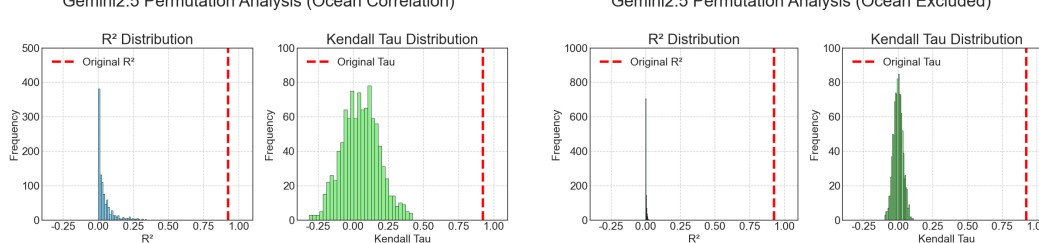

Figure 10: Permutation tests for Gemini 2.5 confirm statistical significance ($p < .001$). The observed statistics (red dashed line) for $R^2$ and Kendall's $\tau$ are extreme outliers relative to the empirical null distribution.

**Robustness to Task Framing.** To ensure the structural amplification effect arises from genuine psychological inference rather than being an artifact of the experimental setup, we tested the model's invariance across three distinct conditions: our original "Standard Order" setup, a "Random Order" setup and a "Single Question" setup.

As shown in Figure 11, the amplification coefficient ($k$) for Gemini 2.5 remained stable across these conditions ($k = 1.42$, 1.41, and 1.42, respectively). This consistency is significant as these conditions systematically varied the task structure. The "Random Order" condition, which shuffled the sequence of the 20 input Big Five items, demonstrates that the model's reasoning is robust against variations in the input information's structure. The "Single Question" condition, which altered the task from predicting an entire questionnaire to predicting each item individually, shows that the effect is also robust against changes in the output task's format. The remarkable stability across these manipulations confirms that structural amplification is a core feature of the model's reasoning process, not a byproduct of a specific input-output format.

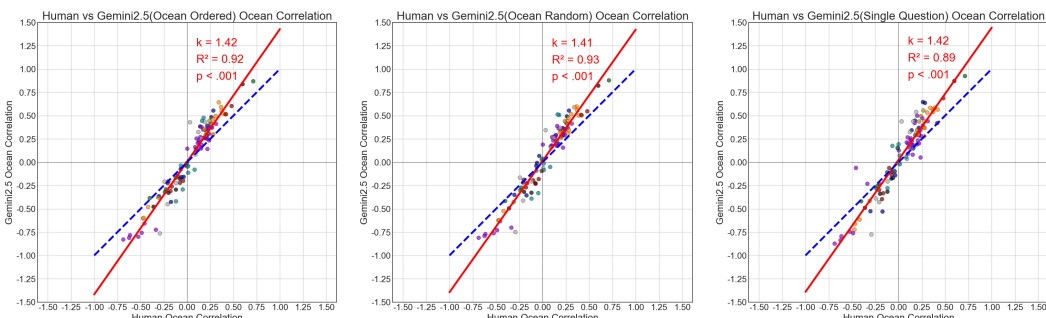

Figure 11: Robustness of the structural amplification effect in Gemini 2.5. The amplification multiplier ($k$) remains highly stable across Standard Order, Random Order and Single Question conditions.

## C.2 Deconstructing the Reasoning Mechanism

### C.2.1 Cross-Model Consensus in Attribution Strategy

To quantitatively assess the consistency of the information selection strategy across different models, we performed a pairwise comparison of all attribution vectors generated. This analysis covered every combination of Reasoning Model and Annotation Model.

As shown in Figure 12, the resulting attribution distributions exhibited remarkable alignment. Excluding the trivial diagonal values, the average Pearson correlation coefficient was exceptionally high at $\rho = 0.9343$, while the average Kullback-Leibler (KL) divergence was extremely low at $D_{KL} = 0.0475$. Together, these metrics confirm that all models independently converge on a near-identical, fundamental attribution strategy.

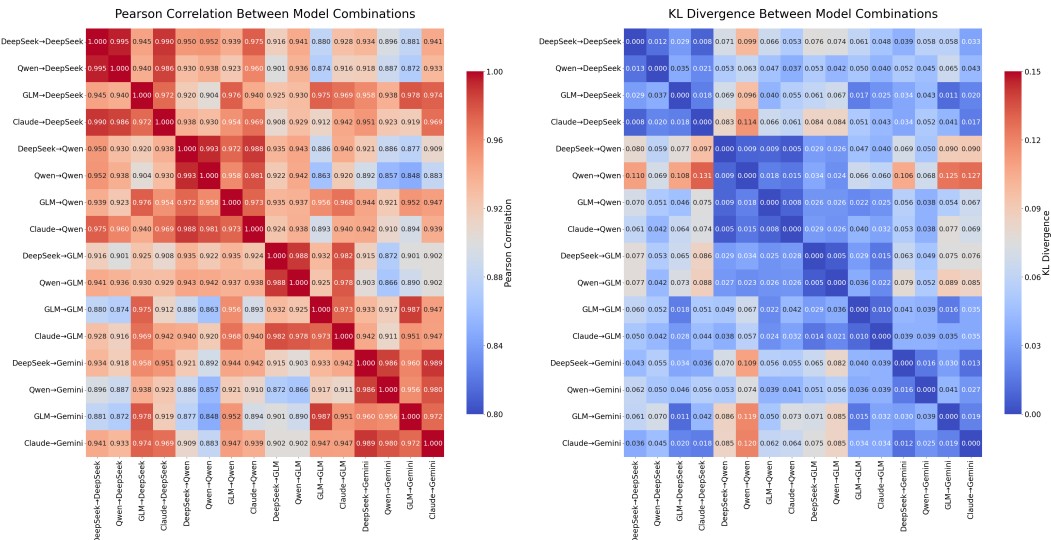

Figure 12: Pairwise comparison of attribution strategies across model combinations. Left: Heatmap of the Pearson correlation ($\rho$). Right: Heatmap of the Kullback-Leibler ($D_{KL}$) divergence. Each axis label follows the format Reasoning Model -> Annotation Model. Together, the high correlations and low divergences demonstrate a strong cross-model consensus.

## C.3 Efficacy and Synergy of Abstract Summaries

To empirically validate our amplification coefficient ($k$), we tested its association with predictive performance, measured by the mean correlation ($r$). The analysis used data from 15 experimental conditions, generated by crossing five large language models (DeepSeek, GLM, Qwen, Claude and Gemini) with three information input scenarios (SummaryOnly, ScoreOnly, and Summary+Score). The raw numerical results for each condition are listed in Table 2.

As shown in a scatter plot of these data points (Figure 13), there is a strong, positive linear relationship between the two metrics. A high coefficient of determination ($R^2 = 0.93$) confirms that variations in amplification account for most of the variance in predictive correlation. This result strongly supports our claim that structural amplification is a key mechanism corresponding directly to enhanced predictive power.

Table 2: Amplification coefficients ($k$) and mean predictive correlations ($r$) for each model across the three information type conditions.

| Model | Information Type | Mean Correlation ($r$) | Amplification Coefficient ($k$) |
|---|---|---|---|
| DeepSeek | SummaryOnly | 0.361 | 1.36 |
| | ScoreOnly | 0.376 | 1.38 |
| | Summary+Score | 0.402 | 1.44 |
| GLM | SummaryOnly | 0.395 | 1.44 |
| | ScoreOnly | 0.418 | 1.52 |
| | Summary+Score | 0.421 | 1.54 |
| Qwen | SummaryOnly | 0.399 | 1.51 |
| | ScoreOnly | 0.432 | 1.56 |
| | Summary+Score | 0.435 | 1.61 |
| Claude | SummaryOnly | 0.403 | 1.54 |
| | ScoreOnly | 0.442 | 1.62 |
| | Summary+Score | 0.445 | 1.65 |
| Gemini | SummaryOnly | 0.345 | 1.33 |
| | ScoreOnly | 0.374 | 1.41 |
| | Summary+Score | 0.389 | 1.44 |

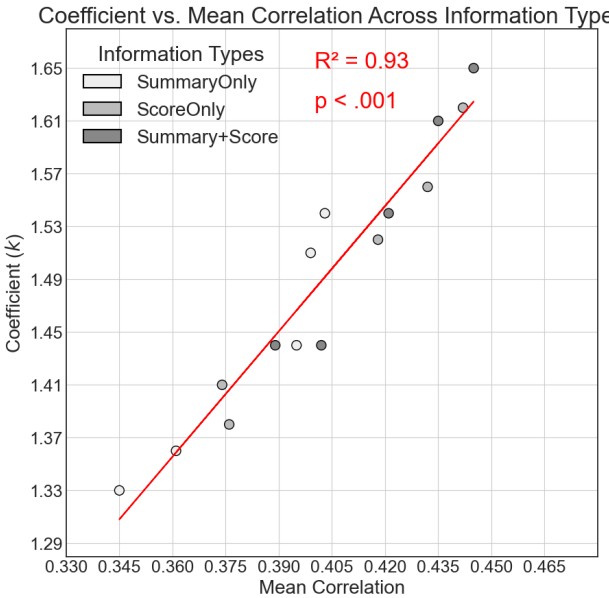

Figure 13: Correlation between the amplification coefficient ($k$) and mean predictive correlation ($r$) across 15 conditions (5 models $\times$ 3 information types). The plot shows a strong, significant positive linear relationship ($R^2 = 0.93, p < .001$), demonstrating that higher structural amplification is associated with improved predictive performance.

## C.4 Idealization Hypothesis: Reliability and Noise Analysis

### C.4.1 Data Reliability and Attenuation Correction

This section provides the detailed results supporting the reliability analysis discussed in the main text. We demonstrate that (1) LLM-generated data has a higher internal consistency than human data, and their reliability profiles converge, and (2) correcting for the attenuation in human data brings the amplification coefficient ($k$) close to 1.0.

Figure 14 shows that LLM-generated data exhibits significantly higher internal consistency (Cronbach's Alpha) than human data across most psychological constructs (specifically, in 16 out of 18 cases; mean $\alpha_{\text{LLM}} = 0.87$ vs. $\alpha_{\text{Human}} = 0.75$). This provides direct evidence that LLMs bypass a significant source of statistical noise inherent in human self-reports, which can stem from factors such as inattention, fluctuating emotional states, or momentary misinterpretation of items. Furthermore, our analysis of the reliability profiles confirms a strong convergence among LLMs towards a shared, idealized response model. The mean Mean Squared Error (MSE) between any two LLM profiles was substantially lower than between any LLM and the human profile (mean inter-LLM MSE = 0.0060 vs. mean LLM-to-Human MSE = 0.0357), highlighting that various LLMs share a similar cognitive pattern that is systematically distinct from human.

Figure 14: Reliability Comparison: Human vs. AI Models Across Psychological Constructs. This bar chart displays the Cronbach's Alpha coefficients for data generated by humans (blue) and the mean across all AI models (purple, with error bars indicating standard deviation). In the majority of cases, LLM-generated data exhibits higher internal consistency, supporting the hypothesis that LLMs filter random measurement error.

To directly test whether the structural amplification phenomenon is an artifact of measurement error, we performed a unilateral correction for attenuation on the human correlation data, as recommended by classical test theory (Nunnally, 1975). This standard psychometric procedure uses the reliability coefficients (Cronbach's Alpha) of two scales to estimate the "true score" correlation between them, effectively removing the attenuating effect of measurement error. We applied this correction to the entire human correlation matrix and then re-calculated the amplification coefficients ($k$) by regressing the original LLM correlation matrices against this newly disattenuated human matrix. As shown in Figure 15, after this correction, the amplification coefficients for all Large Language Models dropped significantly, clustering around a value of 1.0. This result demonstrates that the amplification effect is largely accounted for by the unreliability in human-generated data, and that LLMs are effectively reconstructing a psychological structure akin to the "true score" correlations.

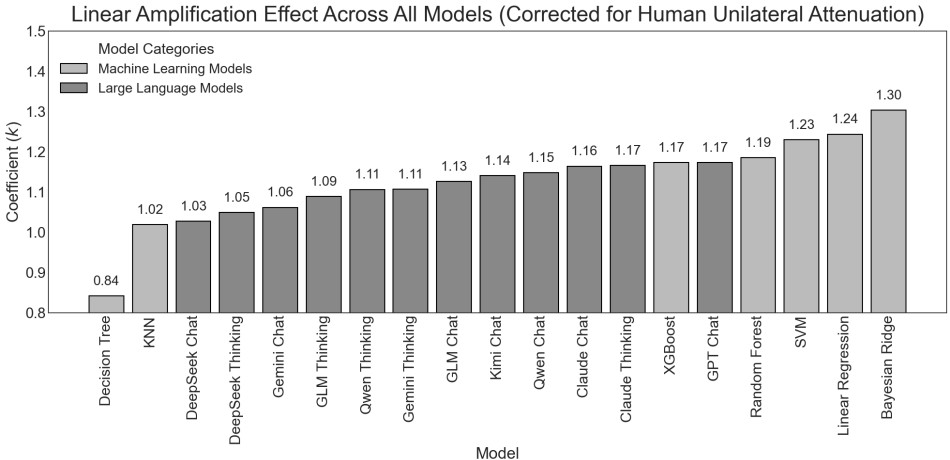

Figure 15: Amplification Coefficients After Unilateral Attenuation Correction. The amplification coefficients ($k$) for all LLMs now cluster around 1.0 after correcting for measurement error in the human data. This suggests they are modeling the disattenuated, "true score" psychological structure, rather than simply amplifying noisy, observed correlations.

### C.4.2 EMPIRICAL SUPPORT FOR THE IDEALIZATION HYPOTHESIS

In Section 4.1, we posit that the LLM may function as an "idealized participant" by abstracting away the noise inherent in human responses. To provide empirical support for this hypothesis, we conducted two complementary analyses on both human and model side.

**Analysis of Attentive Human Participants.** To isolate the effect of human inattention—a key source of data noise—we identified a subgroup of attentive participants ($N_{\text{attentive}} = 309$) from the full dataset ($N = 816$). This was operationalized by excluding individuals based on their response times. Specifically, the attentive subgroup consists of participants whose response times on all questionnaires were consistently above a lower-bound threshold, defined as the mean of inter-response time differences minus half a standard deviation. As shown in Figure 16, this low-noise subgroup exhibited stronger internal correlations, quantified by an amplification multiplier of $k = 1.08$ when compared to the full sample. This provides strong correlational evidence that as human-generated noise decreases, empirical data converges toward the idealized structure captured by the LLM.

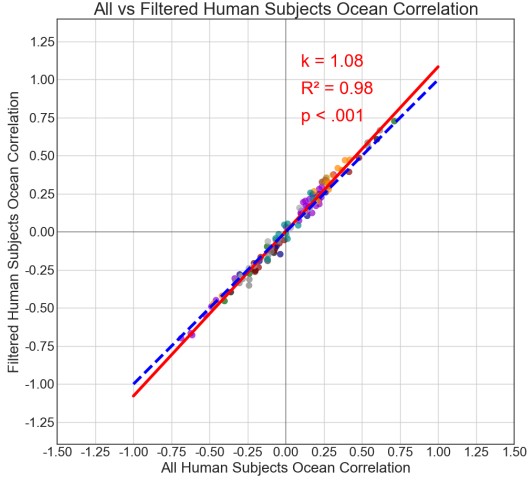

Figure 16: Scatter plot comparing the OCEAN intra-correlation coefficients derived from the full human sample (All Subjects, N=816) against the more attentive subgroup (Filtered Subjects, N=309).

**The Noise Injection Experiment.** To establish a near-causal link, we intervened on a baseline Linear Regression model by systematically adding Gaussian noise ($\sigma \in \{0.25, 0.5, 0.75, 1.0\}$) to its predictions. As visualized in Figure 17, this revealed a clear dose-response relationship: as injected noise increased, the amplification coefficient $k$ systematically decreased from 1.55 to 1.12. This inverse relationship between noise and amplification supports our hypothesis that LLMs achieve this effect through a process of noise filtering.

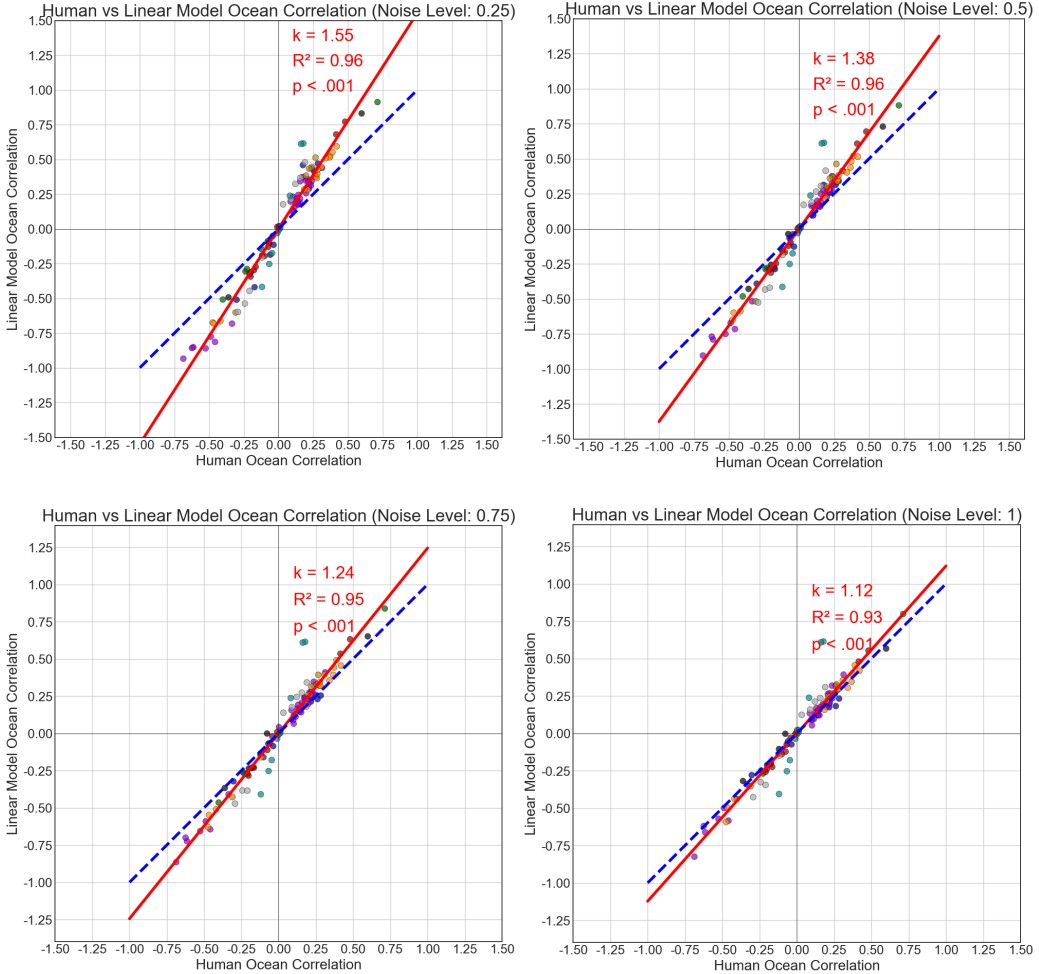

Figure 17: Results of the noise injection experiment. As the level of Gaussian noise added to the linear model's predictions increases from 0.25 to 1.0, the amplification coefficient ($k$) systematically decreases, demonstrating a clear dose-response relationship.

# D  TARGET PSYCHOLOGICAL SCALES

For clarity and reproducibility, the following table lists the full names and abbreviations of the psychological scales used as prediction targets in our experiments. Detailed descriptions of the scales and their items are available in our GitHub repository.

Table 3: List of Sub-scale Abbreviations, Their Parent Scales, and Color Mappings.

| Parent Scale | Sub-scale Abbreviation | Color |
|---|---|---|
| Perceived Stress Scale (Cohen et al., 1983; Leung et al., 2010) | PSS | black |
| Simplified Coping Style Questionnaire (Xie, 1998) | Coping_P
Coping_N | darkblue |
| State-Trait Anxiety Inventory (Trait, Spielberger et al., 1971) | Trait_Anxiety | darkgreen |
| Self-Compassion Scale (Neff, 2003) | Self_Criticism
Self_Isolation
Over_ID
Self_Kindness
Self_Humanity
Self_Mindfulness
Self_Compassion | darkviolet |
| Psychological Resilience Scale (CD-RISC) (Connor & Davidson, 2003) | PR_Tenacity
PR_Strength
PR_Optimism | darkorange |
| Intolerance of Uncertainty Scale (Buhr & Dugas, 2002) | Pros_Anx
Inhib_Anx | darkred |
| Emotion Regulation Questionnaire (Gross & John, 2003) | ERQ_R
ERQ_S | darkgray |
| Risk Perception & Behavior Questionnaire | Risk_Unrel
Obj_Risk | darkcyan |
| Future Time Perspective Scale (Carstensen & Lang, 1996) | FT_Persp | darkgoldenrod |

## LLM USAGE STATEMENT

In this work, large language models served as an assistive tool in the research and writing process. The model's contributions included assisting with drafting and iteratively revising the manuscript—such as generating initial text and rephrasing sentences to improve clarity—and helping to write and debug Python code for the data analysis pipeline. For manuscript presentation, the LLM's role was specifically limited to generating LaTeX code for several icons used in schematic figures and assisting with the aesthetic formatting of tables. The authors retained full intellectual control and bear complete responsibility for the final content, having critically reviewed, edited, and validated all AI-generated contributions. The model is therefore not eligible for authorship, and its role is acknowledged here in the spirit of transparency.

