# OpenReview forum: "From Five Dimensions to Many: Large Language Models as Precise and Interpretable Psychological Profilers"
_ICLR.cc/2026/Conference — ICLR 2026 Poster_

### Official Review · Reviewer_GETG · 2025-10-30

**Soundness:** 4
**Presentation:** 2
**Contribution:** 2
**Rating:** 6
**Confidence:** 4

**Summary:**

This paper proposes a controlled paradigm to test whether large language models can infer psychological structures from the big-5 personality traits of a user. Given individual responses from 20 elements in a big-5 questionnaire, models are prompted to role-play that individual across 9 other, distinct, psychological scales. In the first experiment, the paper compares the inter-scale correlation matrices between. model-generated scores and human ground-truths, showing that LLMs often closely reproduce humans' correlation patterns (Gemini, R=0.92), and that they display an amplification effect in the correlations. In a second experiment, the paper investigates how models achieve this correlation, by focusing on the reasoning traces generated during thinking, showing that models are consistent (there is high correlation between the attribution vectors of different reasoning models), but do not have fine-grained attribution ability (instead focusing on modre global patterns), which is confirmed by ablation (using a summary in addition to the original numeric elements).

**Strengths:**

This is an interesting paper, which introduces a fairly novel way of evaluating how well LLMs can capture and express personalities. The experiments are well thought out, and they show some interesting artifacts, particularly the results on attribution and COT reasoning/summarization. The methodology is strong, and the results are convincing (and likely reproducible). The work is reasonably clear in its core motivation and experimental pipeline, and the finding that LLMs can reconstruct and amplify latent psychological structure from minimal inputs has potential implications for downstream applications (particularly in behavioral simulation)

**Weaknesses:**

- The related work section is somewhat thin on detail, and largely ignores a rich set of works designed to approximate specific behaviors, even up to the Big-5 personality traits. Some relevant related work beyond the park study of LLMs include [1,2,3,4,5,6,7,8,9]. It's really not clear to me that the paper significantly differentiates itself solely by " introducing a well-controlled psychometric paradigm", since many of these other paper do the same (either in the domain of behavior, or opinion). There are also papers which use Big-5 as conditioning, such as [13,14], and the ability of LLMs to emulate personality was explored in [15] (which is quite similar to this work).
- There's no ablation of the experimental prompt in A.1. It's likely that using a system paradigm, with the response in the user, compared to several back to back user turns will affect the performance of the model (and the strength of the correlations). Similarly, it's likely that the wording of the system prompt has impacts on the output. It would be good to perform some analysis accounting for that. In addition, it's interesting that all models here are instruction-tuned models: how does this change if a base model is used instead?
- Some of the results seem a bit cherry-picked per-model/experiment. For example, Figure 2 only discusses Gemini, but how does this change for other models?

More minor issues:
- The scales used are never discussed in the main text, and only included in Appendix D.
- Language models are only discussed in Appendix E (which makes it really hard to follow what the overall experiments are).
- I'm not entirely sure how novel the "amplifcation" idea is. Several papers [10,11,12] have explored how LLMs amplify biases (and much of this tracks with how LLMs perform maximum likelihood prediction).
- The presentation of the paper is a bit dense, and wanders a lot, making it hard to understand exactly what is being tested, what the research questions are, and what experiments are being performed on which models. Restructuring the paper around clear RQs and hypotheses would significantly improve the readability of the paper.
- There's no pipeline ablation in this paper (ie. the annotation model in Figure 3), which makes it hard to understand if the underlying results are from biases in the annotation models, or biases in the data itself.

[1] Ziems, Caleb, et al. "Can large language models transform computational social science?." Computational Linguistics 50.1 (2024): 237-291.
[2] Dillion, Danica, et al. "Can AI language models replace human participants?." Trends in Cognitive Sciences 27.7 (2023): 597-600.
[3] Aher, Gati V., Rosa I. Arriaga, and Adam Tauman Kalai. "Using large language models to simulate multiple humans and replicate human subject studies." International conference on machine learning. PMLR, 2023.
[4] Tjuatja, Lindia, et al. "Do llms exhibit human-like response biases? a case study in survey design." Transactions of the Association for Computational Linguistics 12 (2024): 1011-1026.
[5] Choi, Hyeong Kyu, and Yixuan Li. "Picle: Eliciting diverse behaviors from large language models with persona in-context learning." arXiv preprint arXiv:2405.02501 (2024).
[6] Hilliard, Airlie, et al. "Eliciting personality traits in large language models." arXiv preprint arXiv:2402.08341 (2024).
[7] Santurkar, Shibani, et al. "Whose opinions do language models reflect?." International Conference on Machine Learning. PMLR, 2023.
[8] Kang, Minwoo, et al. "Deep Binding of Language Model Virtual Personas: a Study on Approximating Political Partisan Misperceptions." Second Conference on Language Modeling.
[9] Suh, Joseph, et al. "Rediscovering the Latent Dimensions of Personality with Large Language Models as Trait Descriptors." NeurIPS 2024 Workshop on Behavioral Machine Learning.
[10] Taori, Rohan, and Tatsunori Hashimoto. "Data feedback loops: Model-driven amplification of dataset biases." International Conference on Machine Learning. PMLR, 2023.
[11] Wang, Ze, et al. "Bias Amplification: Large Language Models as Increasingly Biased Media." arXiv preprint arXiv:2410.15234 (2024).
[12] Peña Fernández, Simón, et al. "Without journalists, there is no journalism: the social dimension of generative artificial intelligence in the media." Profesional de la información 32.2 (2023).
[13] Li, Wenkai, et al. "Big5-chat: Shaping llm personalities through training on human-grounded data." arXiv preprint arXiv:2410.16491 (2024).
[14] Vu, Huy, et al. "Psychadapter: Adapting llm transformers to reflect traits, personality and mental health." arXiv preprint arXiv:2412.16882 (2024).
[15] Wang, Yilei, et al. "Evaluating the ability of large language models to emulate personality." Scientific reports 15.1 (2025): 519.

**Questions:**

- How does this change for non instruction-tuned models? it seems like many of these artifacts may come from the optimization process during RLHF or similar tuning processes.
- Is there a correlation between length of the thinking trace, and some of the "summary" behavior? It feels like in some cases, the thinking trace performs as a summary for the model, allowing it to reason/perform test-time compute instead of directly producing an output.

---

> ### Author Response · Authors · 2025-11-21
> **Response to Reviewer GETG (Part 1/2)**
>
> We are very grateful to the reviewer for their exceptionally thorough evaluation and for recognizing the soundness of our methodology. Your detailed feedback, particularly regarding the paper's presentation and its positioning within the literature, has been invaluable. We have undertaken a significant revision based on your suggestions and believe the paper is much stronger and clearer as a result.
>
> Below, we address each of the weaknesses and questions in detail.
>
> ### **Response to Weaknesses**
>
> 1. **On the Related Work Section:**
>    Thank you for this crucial feedback and for providing an extensive list of relevant literature. We agree that our original section was insufficient. In the Related Work section (Section 2) of our revised manuscript, we now incorporate all suggested references to better situate our work.
>
>    Regarding our paper's differentiation, while some prior work uses the Big Five as a condition [13,14] or evaluates personality emulation [15], our work is distinct in three key ways, which we now highlight more clearly:
>
>   * **Second-Order Analysis:** We move beyond first-order trait prediction to a second-order analysis of the entire correlational network between traits, assessing the structural fidelity of the model's psychological reasoning.
>   * **Reasoning Trace Analysis:** We analyze the models' spontaneously generated reasoning traces to understand how they arrive at their predictions, moving beyond performance metrics to explanatory analysis.
>   * **Breadth of Scales and Network-Level Analysis:** Our methodology leverages the breadth of the nine target scales to conduct a more holistic, network-level analysis. Instead of only examining the correlations between the input (Big Five) and individual predicted scales, our analysis evaluates the full matrix of inter-correlations across all psychological constructs involved. This comprehensive approach allows us to assess the LLM's ability to reconstruct the entire psychological network of traits, moving beyond isolated input-output predictions.
>
> 2. **On Ablation of the Experimental Prompt & Use of Base Models:**
>
>    This is an important point. We address the two issues raised:
>
> * **Prompt Sensitivity:** We appreciate the suggestion to test prompt sensitivity. To address this, we conducted a robustness analysis, now detailed in Section 3.3, where we tested three distinct conditions: "Standard Order", "Random Order" (shuffling input items), and "Single Question". The stability of our results across these conditions demonstrates that the amplification effect is a robust feature of the model's reasoning, not an artifact of a specific prompt structure. We also clarify that our experiments use single-turn interactions, mitigating concerns about multi-turn conversational effects.
> * **Instruction-Tuned vs. Base Models:** This is an important question, and we appreciate the opportunity to elaborate on this critical methodological choice. We opted not to use base models for two primary reasons:
>
>   1. **Fundamental Mismatch in Objectives:** Base models are trained purely on a next-word prediction task. Our experiment, however, requires the model to understand and follow complex instructions, including role-playing a specific persona and adhering to a structured output format. The pre-training objective does not equip models with this necessary level of instruction-following capability.
>   2. **Known Reliability Issues:** More critically, pre-trained base models are known to have several well-documented issues that would compromise the reliability of our findings. They often suffer from significant hallucination, a lack of truthfulness, and can generate toxic text (Ouyang et al., 2022; Lin et al., 2022; Ji et al., 2023). Attempting to draw conclusions about nuanced psychological reasoning from such unreliable outputs would be methodologically unsound.
>
>      Ouyang, L., Wu, J., Jiang, X., Almeida, D., Wainwright, C., Mishkin, P., ... & Lowe, R. (2022). Training language models to follow instructions with human feedback. Advances in neural information processing systems, 35, 27730-27744.
>
>      Lin, S., Hilton, J., & Evans, O. (2022, May). Truthfulqa: Measuring how models mimic human falsehoods. In Proceedings of the 60th annual meeting of the association for computational linguistics (volume 1: long papers) (pp. 3214-3252).
>
>      Ji, Z., Lee, N., Frieske, R., Yu, T., Su, D., Xu, Y., ... & Fung, P. (2023). Survey of hallucination in natural language generation. ACM computing surveys, 55(12), 1-38.
>
>      Therefore, to ensure the validity and reliability of our results, we adhered to   established practices by conducting our experiments on instruction-tuned models.

---

> ### Author Response · Authors · 2025-11-21
> **Response to Reviewer GETG (Part 2/2)**
>
> 3. **On Potentially Cherry-Picked Results (Figure 2):**
>    We apologize for not having provided scatter plots for all the models in our previous manuscript.  We had showed results only for Gemini in the original Figure 2 just to save presentation space. To provide a complete picture, we have now added a new section in the Appendix (Appendix B.2) containing the corresponding scatter plots for all 12 LLMs evaluated in our study, demonstrating that the structural amplification phenomenon is a general property across models.
> 4. **On Missing Details in the Main Text (Scales and Models):**
>    Thank you for pointing this out. The lack of these details in the main text made the paper difficult to follow. To improve readability, we have now added the psychological scales and the models used directly into the main experimental section.
> 5. **On the Novelty of "Amplification":**
>    We thank the reviewer for these references [10,11,12], which helped us sharpen our definition. In the revised manuscript (Section 3.2), when we first introduce "structural amplification", we now explicitly contrast it with the more common "bias amplification" to clarify its novelty as a second-order, network-level phenomenon rather than the exaggeration of a specific, first-order bias.
> 6. **On the Presentation of the Paper:**
>    We agree with the reviewer that the original presentation was dense and could have been clearer. We have restructured the paper to be organized around clearer research questions and hypotheses, which we hope significantly improves its readability.
> 7. **On Ablation of the Annotation Pipeline:**
>    This is an excellent point and was an oversight in our original description. To mitigate the bias of any single annotation model, we actually used a diverse suite of four LLMs for the annotation task and averaged their outputs to create a robust, de-biased attribution vector. We have now explicitly stated this methodology in Section 4.1 to make our debiasing strategy clear.
>
> ### **Response to Questions**
>
> 1. **On Non-Instruction-Tuned Models:**
>     As addressed in the weaknesses section, our decision was based on ensuring the models could reliably follow the task instructions and produce results that were not confounded by issues like hallucination common in base models. The phenomena we observed are indeed likely related to the fine-tuning process, which is part of what makes these models capable of such complex inference in the first place.
> 2. **On the Correlation Between Trace Length and Summary Behavior:**
>     This is a very interesting hypothesis. We analyzed the length of the generated summaries and found them to be quite short, with the longest summary containing only 398 words. This is well within the standard context length of modern LLMs (e.g., 4096 tokens) and suggests the summary functions as a compressed, short-text representation of the user's personality, rather than a mechanism for extended "test-time compute" via a long context window.
>
> Once again, we wish to express our sincere appreciation for your rigorous and constructive feedback. Your insights have been critical in helping us refine the paper's structure, positioning, and clarity. We hope our revisions have successfully addressed your concerns.

---

### Official Review · Reviewer_UCxr · 2025-11-01

**Soundness:** 2
**Presentation:** 3
**Contribution:** 3
**Rating:** 8
**Confidence:** 4

**Summary:**

The paper presents a study of the ability of LLMs using chain-of-thought prompting to reconstruct personality profiles using psychometric data and use this information to reconstruct scores on other psychometric measures in a coherent and consistent manner compared to the original human behavioral scores. By analyzing reasoning traces, the authors find that LLMs are reasoning from representations and not simply performing semantic pattern matching.

**Strengths:**

The paper is well-argued and well-structured. By comparing scores across a range of psychometric tests, the experimental design address a fundamental gap in the literature, namely with regard to the coherence and stability of LLM personalities. Further, the experimental setup allows for comparisons across models and across learning algorithms, which strengthens the findings.
The paper offers an insightful analysis and discussion of LLMs tendency towards idealized representations.

**Weaknesses:**

The correlational analysis of Experiment 1 presented in 3.2 is interesting from an exploratory perspective but it is not very convincing and slightly problematic: Multiple comparisons are discouraged and should be replaced with more robust multivariate tests. Indeed, experiment 2 and the subsequent analysis in 4.2 is much more conclusive.

**Questions:**

P30L130: " Our findings chart a path up this hierarchy, demonstrating that LLMs’ cognitive process (1) transcends surface-level statistical association, (2) prioritizes abstract conceptual structure over specific item-level details, and (3) performs a novel form of theoretical idealization, actively refining noisy inputs into theory-consistent representations"
=> It's not immediately clear how a correlational study, even a sophisticated one, can demonstrate that LLMs prioritize abstract conceptual structure.The experimental setup prompts chain-of-thought reasoning. Hence, how is any evidence of abstract reasoning not simply an artifact of the prompt structure?

P5-6L264-276: "We performed two further analyses to validate the observed linear structural amplification is genuine and robust. First, we established statistical significance using a 1,000-trial permutation test. [...] Second, to verify the effect arises from genuine understanding rather than semantic mapping, we examined the model’s sensitivity to task framing or input ordering (Oren et al., 2023) by testing three conditions: our original setup (Standard Order), a setup with shuffled input items (Random Order), and a setup where each prediction was made individually (Single Question"
=> This is good for excluding context effects like chat history, but I fail to see how this validates that the amplification is not a result of chain-of-thought prompting.

P7L344-247: "This clear dissociation uncovers a key insight into the models’ reasoning: LLMs robustly identify the correct personality factor (e.g., Neuroticism) but struggle to differentiate the importance of specific items within that factor. This supports the hypothesis of a top-down, concept-driven process where high-level abstract knowledge guides the inference, even if low-level execution is imprecise."
=> Or the abstraction is belied by the lack of internal representation. This confusion could equally result from semantic pattern matching. How do you exclude this possibility? The high levels of noise suggest that the regression models are incomplete. Did you prompt the models to explain their choices?

---

> ### Author Response · Authors · 2025-11-21
> **Response to Reviewer UCxr (Part 1/2)**
>
> We sincerely thank the reviewer for their thorough review and supportive rating. We are pleased they found the paper "well-argued and well-structured". Your feedback has highlighted a key area for clarification regarding our experimental setup, which we address first, before responding to the other valuable points raised.
>
> First, we would like to clarify a central point regarding our experimental setup, as it appears to be the root of several of the reviewer's valid questions. Our study did not use Chain-of-Thought (CoT) prompting for the role-playing task. We did not include phrases like "let's think step by step" or other strategies to explicitly elicit a reasoning process. The prompts consisted of simple task instructions for role-playing, such as ”Please imagine you are role-playing a specific person.” We did, however, evaluate models that are capable of reasoning and analyzed the reasoning traces they spontaneously produced. We apologize for the confusion.
>
> **On the Weakness of Correlational Analysis in Experiment 1**
>
> We agree that focusing on a large number of individual comparisons can be problematic. In response, we have revised our analysis in two ways:
>
> 1. We have removed the statistical significance markers from the heatmap in Experiment 1 to shift the focus away from individual p-values and toward a more holistic comparison of the correlation patterns.
> 2. To provide a more robust statistical foundation, we have incorporated a reliability analysis (using Cronbach's Alpha), detailed in Section 3.4 and Appendix C.4.1. This analysis provides stronger evidence for our claim that LLMs produce more internally consistent and "idealized" responses than human participants.

---

> ### Author Response · Authors · 2025-11-21
> **Response to Reviewer UCxr (Part 2/2)**
>
> **Response to Questions:**
>
> 1.  **On Demonstrating Prioritization of Abstract Structure & Prompting Artifacts:**
>
>     As clarified above, since we did not use CoT prompting, the abstract reasoning we observe cannot be an artifact of such a prompt structure. The model spontaneously generated high-level conceptual summaries from the raw item-level data, which we believe is strong evidence of it prioritizing abstract structures.
>
>     Furthermore, our robustness checks (Standard Order, Random Order, Single Question) were designed to rule out artifacts related to input sequencing. As demonstrated by Oren et al. (2023), sensitivity to input order can be an indicator of a form of data contamination, where models may have memorized specific training or benchmark sequences. Our findings—that the results are stable even when the input order is shuffled—confirm that the model's reasoning is not dependent on a specific, potentially memorized, input sequence.
>
> 2.  **On Semantic Pattern Matching vs. Abstraction:**
>
>     This is an excellent question, raising a critical point about alternative explanations. We address your points regarding semantic pattern matching, the “high levels of noise”, and model explanation below.
>
>     *   **Excluding Semantic Pattern Matching:** To explicitly rule out the possibility that the observed abstraction is merely sophisticated semantic pattern matching, we have introduced a crucial baseline comparison in the revised manuscript. In Section 4.1, we benchmarked the LLMs against a Semantic Similarity model, which operates purely on the surface-level semantic overlap between input and target items. As shown in Figure 4, the LLMs' reasoning aligns more closely with human-like conceptual factors than with surface-level semantics. This provides strong empirical evidence that the models' cognitive process transcends simple semantic matching.
>
>     *   **Clarifying the "High Levels of Noise":** We sincerely apologize for the confusion caused by our previous wording. We acknowledge that our original phrasing, which may have included terms like "noisy alignment" at the item level, was imprecise. We recognize that this "noise" or "confusion" at the item-level does not necessarily imply a lack of internal representation, but rather a different level of processing. We have revised our language in Section 4.1 to more accurately describe this finding: models exhibit Factor-Level Accuracy Despite Item-Level Confusion. This new phrasing clarifies that models prioritize high-level, abstract factors over the specific weights of individual items, which is precisely what supports our hypothesis of a top-down, concept-driven process.
>
>     *   **On Models Explaining Their Choices:** You asked if we prompted the models to explain their choices. While we did not use a separate prompt to ask for post-hoc explanations, we directly addressed this by analyzing the spontaneously generated reasoning traces from our "Thinking" models. The "explanation" for their choices is precisely what we extracted from these traces. The entire analysis in Section 4.1, which maps the model's final prediction back to the specific Big Five input items it considered, is our method for uncovering and quantifying the model's decision-making process. This provides a direct, in-process window into their choices, rather than relying on a separate, potentially unfaithful, explanatory step.
>
> In closing, we wish to thank the reviewer again for their constructive feedback. Your questions were crucial in helping us recognize and clarify a central point about our methods, which has made the paper significantly clearer and more robust.

---

### Official Review · Reviewer_Pih4 · 2025-11-01

**Soundness:** 2
**Presentation:** 2
**Contribution:** 2
**Rating:** 4
**Confidence:** 2

**Summary:**

The paper studies whether LLMs can recover the latent structure of personality. (1) Given only each person’s Big Five item responses, models ‘role-play’ the person on other 9 psychological scales, and the resulting correlation structure closely mirrors human data but with a consistent, stronger signal. (2) Analysis of generated reasoning traces suggest a two-stage process: compressing human responses into summaries, then reasoning from those summaries to answer new items. Controls rule out simple semantic matching and indicate the effect isn’t due to chance. The authors argue the amplification reflects “idealized version” of human participants, with LLMs filtering human noise.

**Strengths:**

- Moving from the first‑order prediction to the second‑order correlation‑structure analysis is an interesting reframing of the psychology questions with LLMs. The heatmaps and regression plots in Figure 2 (page 5) compellingly visualize the phenomenon.

- An attempt to mechanism‑seeking analysis: factor‑level and item‑level attribution comparison in Figure 4, and amplification depending on information type in Figure 5 are good steps toward understanding the model's decisions rather than reporting raw scores alone.

**Weaknesses:**

- Related Work (Section 2) significantly lacks depth to situate the paper within existing literature.

In particular, it overlooks a growing body of research suggesting that LLMs exhibit latent psychological structure of humans. Few papers I found by search are “Personallm: Investigating the ability of gpt-3.5 to express personality traits and gender differences” (Jiang et al.), “Personality traits in large language models” (Serapio-Garcia et al.), “Rediscovering the Latent Dimensions of Personality with Large Language Models as Trait Descriptors” (Suh et al.), to name a few. Currently the related work is a bland listing of a few disconnected sets of papers, and must be strengthened to understand the value of this paper.

- LLMs, especially reasoning models, would have utilized its ‘knowledge’ of correlation between traits rather than role-playing.

Although inputs to LLM are just Big-Five item results, LLM still reads what are the target items it will predict (e.g., “ERQ questionnaire items” in Line 782). Their well‑known semantics (i.e., LLMs know what is ERQ, and possibly how it is related to Big-Five items from pretraining corpus and past psychological research results) could let the model rely on learned associations between Big-Five traits and those constructs rather than inferring a structure unique to this dataset. I think this is especially prevalent when using reasoning models. Reasoning models tend to produce reasoning trajectories in terms from a third-person perspective rather than actual role-playing a given persona / traits, try to come up with logical explanations, far from what the paper claims LLMs do to make a prediction. However, from the current paper I cannot see how the model actually works nor can run reproducible experiments; it would be great to see the real example of Figure 3 (“reasoning-to-annotation” analyses).

- Is ‘linear amplification’ an ideal thing, and isn’t it confounded with unreliability (Section 3)?

The slope k>1 is interpreted as “idealization”. However, when the human correlation matrix is attenuated by measurement variance and inherent variability during the human's choice making process, while the LLM prediction correlation matrix is less noisy, regressing the latter on the former will naturally produce a slope k>1. Also, internal consistency (measured in terms of Cronbach’s alpha, for example) of too high values is not necessarily desirable. Without disattenuating human correlations for scale reliabilities (e.g., Cronbach’s alpha per subscale) or applying classical correction for attenuation, the amplification claim risks restating a reliability artifact.

- Reproducibility.

I checked the repo, and there are only .pdf files of questionnaires and no code available for analysis and experiment, limiting the reproducibility of the experiment since authors have brought multiple ML models for evaluation (Figure 2).

**Questions:**

There are multiple vague terminologies that I ask authors to clarify.

- Line 58 “the test should enable mechanistic interpretability”, line 75 “to further achieve interpretability”, line line 464 “using advanced mechanistic interpretability” -> I don’t get why authors mention mechanistic interpretability. This paper is completely irrelevant to mechanistic interpretability. Could authors please describe how their test enables mechanistic interpretability, e.g. is it able to establish casual relationships through interventions on activations?

- Line 25: “analogous to generating sufficient statistics” -> How is it analogous to sufficient statistics? In the paper, there is no description why and how generating natural language personality summaries is relevant to sufficient statistics..

- Line 193: “linear amplification” -> is linear amplification a terminology used in the community? I think the analysis done here is effectively summarized as calculating correlation coefficients between entries in LLM-prediction correlation matrix and human correlation matrix, getting the coefficient > 1.

- Line 151: “ops‑bge‑reranker‑larger” -> there is no model named “ ops-bge-reranker-larger” exists, looks like a hallucinated model label.

I would like to ask authors about representativeness, availability, and ethical approval detail of 816 human subjects.

- Representativeness: is it a representative sample of the entire population?

- Availability: It “was collected during COVID-19 as a part of a larger study” (Line 143), is it published somewhere as a publicly accessible dataset? If so, please mention the original data source rather than just a description of it. If not, please mention the future availability of it for reproducibility.

- Ethical approval detail: following ICLR code of ethics, I would like to ask authors for more details of human subject recruitments, including IRB process, documentation, etc.

---

> ### Author Response · Authors · 2025-11-21
> **Response to Reviewer Pih4 (Part 1/2)**
>
> We sincerely thank the reviewer for the detailed and critical feedback. The specific points raised are insightful and have been instrumental in helping us substantially improve the clarity, depth, and robustness of our manuscript. We have revised the paper accordingly and truly believe it is much stronger as a result of this review.
>
> Below, we address each of the weaknesses and questions in detail.
>
>
> ### **Response to Weaknesses**
>
> **1. On the Lack of Depth in Related Work**
>
> We thank the reviewer for this important critique and for providing several highly relevant papers. We agree that our original Related Work section was insufficient. In our revised manuscript, we have expanded Introduction and Related Work to not only incorporate the suggested works by Jiang et al., Serapio-Garcia et al., and Suh et al. but also to better situate our contribution within the broader literature on LLMs and personality simulation. This provides a clearer context for our second-order analysis and highlights its novelty.
>
> **2. On LLM Knowledge vs. Role-Playing**
>
> This is a critical point that revealed a lack of clarity in our original paper. We offer three points of clarification:
>
> * **Clarification on Prompts:** We thank the reviewer for pointing out Line 782 regarding "ERQ questionnaire items". This highlights a significant misunderstanding we failed to prevent. The prompt mentioned was given to our secondary annotation model in Experiment 2, which was tasked with parsing the reasoning traces. In our main experiment, the predictive models were never shown the names of the target scales (e.g., "ERQ"). They were only given the raw items to predict, preventing them from directly leveraging pre-trained knowledge about specific psychological constructs.
> * **Ruling out Semantic Associations:** We share the concern that models might rely on simple semantic associations. To explicitly address this, we added an analysis in Section 4.1 benchmarking our LLMs against a Semantic Similarity baseline model. As shown in Figure 4, the LLMs' reasoning aligns more closely with human-like conceptual factors than with surface-level semantics, empirically demonstrating that their performance transcends simple pattern matching.
> * **Availability of Reasoning Traces:** The reviewer requested real examples from our "reasoning-to-annotation" analysis. The example in Figure 3 is a real, representative case. To provide full transparency, we have now uploaded all raw LLM outputs and their complete reasoning traces to our anonymized GitHub repository.
>
> **3. On 'Linear Amplification' vs. a Reliability Artifact**
>
> This is an excellent and critical insight, and we are grateful to the reviewer for raising it. We agree that without accounting for the reliability of human measurements, our "amplification" claim could simply be an artifact of comparing a noisy human signal to a cleaner model signal.
>
> To address this directly, we have conducted a new reliability analysis as suggested. In the revised Section 3.4 and Appendix C.4.1, we now:
>
> 1. Calculate the internal consistency (Cronbach's Alpha) for both the human- and LLM-generated data.
> 2. Perform a correction for attenuation on the human correlation matrix, as prescribed by classical test theory, to estimate the "true score" correlations.
>
> Our primary finding from this new analysis is that LLM-generated data exhibits significantly higher internal consistency than human-generated data across most psychological constructs (mean $\alpha_{\text{LLM}} = 0.87$ vs. $\alpha_{\text{Human}} = 0.75$), as detailed in the new Figure 14. This directly supports our "idealized participant" hypothesis by showing that LLMs produce less noisy and more consistent responses, effectively filtering the random measurement error present in human self-reports.
>
> Furthermore, as a consequence of this reliability difference, when we perform the correction for attenuation you suggested, the amplification coefficient ($k$) indeed moves substantially closer to 1.0 for all LLMs (Figure 15). This aligns with your insight and further strengthens our interpretation that structural amplification stems from models reconstructing a psychological structure closer to its "true score" form, rather than being an arbitrary amplification.
>
> **4. On Reproducibility**
>
> We apologize that the repository was incomplete. To fully address this, we have now uploaded:
>
> * The fully anonymized raw data for all 816 human participants.
> * All psychological questionnaires used.
> * The complete, raw outputs and reasoning traces from all LLM experiments.
>
> We are committed to full reproducibility and will upload the complete analysis code upon acceptance of the paper. To further protect human participants’ privacy, we have anonymized their data using a shuffling procedure on participant identifiers to decouple it from any other potential data sources.

---

> ### Author Response · Authors · 2025-11-21
> **Response to Reviewer Pih4 (Part 2/2)**
>
> ### **Response to Questions**
>
> **1. Clarification of "Mechanistic Interpretability"**
>
> We agree that our use of "mechanistic interpretability" was imprecise. In the AI safety community, this term often implies causal interventions on model internals (e.g., activations), which our study does not perform. We have revised this terminology throughout the paper to be more accurate. Our intent was to move beyond performance metrics to explain how the model arrives at its predictions. We believe the attribution analysis in Section 4.1, which maps model outputs back to specific input items, is a step towards explaining the model's decision-making process, even if it is not "mechanistic" in the strictest sense.
>
> **2. Analogy to "Sufficient Statistics"**
>
> Thank you for requesting clarification. We agree entirely with your point. The term "sufficient statistics" carries a strict statistical meaning, and our initial use of it as an analogy was imprecise and potentially misleading, especially since our own findings in the Summary+Score condition show that the original scores still provide additional predictive value.
>
> In our revised manuscript, we have removed this phrase entirely to avoid any potential confusion with its strict statistical meaning. We appreciate you pointing this out, as the change improves the clarity and precision of our abstract.
>
> **3. Definition of "Linear Amplification"**
>
> In the revised Section 3.2, we now explicitly define "linear amplification" and contrast it with the more common term "bias amplification". While bias amplification typically refers to the exaggeration of a specific, pre-existing societal bias (a first-order effect), our "linear amplification" describes a systematic increase in the magnitude of the entire correlational network (a second-order effect).
>
> However, we agree with your sentiment that our original term could be clearer. To enhance clarity, better distinguish our second-order finding from prior work, and improve understanding, we have now unified the terminology throughout the manuscript to "structural amplification". We believe this term more accurately captures the phenomenon we observed: a systematic amplification of the entire correlational structure of the psychological network.
>
> **4. Clarification of the "ops-bge-reranker-larger" Model**
>
> We apologize for the confusion; this is a real, not a hallucinated, model label. The name refers to a BGE-series cross-encoder model provided via Alibaba Cloud's OpenSearch (ops) service. It is documented on both the [BGE model's official page](https://bge-model.com/tutorial/5_Reranking/5.2.html) and [Aliyun's documentation](https://help.aliyun.com/zh/open-search/search-platform/developer-reference/ranker-api-details). We have clarified this in the revised Section 3.
>
> **5. On Representativeness, Availability, and Ethical Approval of Human Subject Data**
>
> We thank the reviewer for raising these critical ethical questions.
>
> * **Representativeness:** The sample consists of 816 Chinese participants (564 female). The age range is 18-52 (Mean = 25.0, SD = 7.4). Participants were recruited from 32 of China's 34 provinces. Education levels were: Primary or below (3), Junior high (15), High/vocational (43), Bachelor's (573), Master's (157), and Doctoral (25). While geographically broad, we acknowledge in the paper that it is not a fully representative sample of the entire population.
> * **Availability:** The anonymized dataset is now available in our GitHub repository.
> * **Ethical Approval:** This research was conducted with strict adherence to ethical principles. The data was collected as part of a larger study that received full IRB approval. All participants provided informed consent, and all data was fully anonymized before we accessed it. We have clarified the ethical approval issue in the Ethics Statement of the revised manuscript. Due to the double-blind review process, we are currently unable to disclose specific institutional details of the IRB approval, but we are committed to providing this information upon acceptance of the paper.
>
> Once again, we thank you for your rigorous and constructive review. We hope our detailed responses and extensive revisions have adequately addressed your concerns and demonstrated the value of our work.

---

### Official Review · Reviewer_GtxT · 2025-11-01

**Soundness:** 3
**Presentation:** 2
**Contribution:** 3
**Rating:** 8
**Confidence:** 4

**Summary:**

The authors show good alignment between LLM predictions and human psychological structure comparing inter-scale correlation LLMs patterns between LLM predictions and human data. This is stark contrast with recent results by Zhu et al. (2025) showing poor LLM alignment when inferring personality of a narrator from a given text.

The authors claim that the reason of this contrast is that though LLMs struggle to predict specific traits they tend to amplify the correlations between scores on different scales that human subjects show.

This contradiction is particularly interesting, in my opinion, and might span a very fruitful line of further research.

**Strengths:**

The authors suggest a great research question, find an answer to it and demonstrate the validity of the answer in a convincing way.

**Weaknesses:**

I am not an expert in classical psychology literature, but some of the human-score correlations should be taken with a grain of salt due to the fact that they are often reported on small selection of subjects that are not representing the scale of human experience that an LLM is exposed to. In this context the stronger correlation that the authors report is even more surprizing to me, especially in the context of the later remark on more "attentive" subjects.

**Questions:**

Could the correlation amplification phenomenon you find be the result of post-training in the alignment phase, when models are trained to simulate an agreeable personality?



Here are some further works that could be informative for the scope of the work:

Sorokovikova A, Rezagholi S, Fedorova N, Yamshchikov IP. LLMs Simulate Big5 Personality Traits: Further Evidence. InProceedings of the 1st Workshop on Personalization of Generative AI Systems (PERSONALIZE 2024) 2024 Mar (pp. 83-87).

Pan X, Gao D, Xie Y, Chen Y, Wei Z, Li Y, Ding B, Wen JR, Zhou J. Very large-scale multi-agent simulation in agentscope. arXiv preprint arXiv:2407.17789. 2024 Jul 25.

Dong W, Zhao Y, Sun Z, Liu Y, Peng Z, Zheng J, Zhang Z, Zhang Z, Wu J, Wang R, Xu S. Humanizing llms: A survey of psychological measurements with tools, datasets, and human-agent applications. arXiv preprint arXiv:2505.00049. 2025 Apr 30.

Tshimula JM, Nkashama DJ, Muabila JT, Galekwa RM, Kanda H, Dialufuma MV, Didier MM, Kalonji K, Mundele S, Lenye PK, Basele TW. Psychological Profiling in Cybersecurity: A Look at LLMs and Psycholinguistic Features. InInternational Conference on Web Information Systems Engineering 2024 Dec 2 (pp. 378-393). Singapore: Springer Nature Singapore.

---

> ### Author Response · Authors · 2025-11-21
> **Response to Reviewer GtxT (Part 1/2)**
>
> We are very grateful for the reviewer's insightful comments, constructive questions, and positive assessment of our work. We are particularly encouraged by the high rating.  We have incorporated the suggested changes and address each point below.
>
> **1. On the Representativeness of the Human Subject Sample:**
>
> We thank the reviewer for this important point regarding the representativeness of the human subject data, especially in the context of the vast data LLMs are exposed to. We have acknowledged this as a limitation in the Discussion section of our paper and are planning to collect data from a more diverse and larger sample in our future work.
>
> At the same time, we would like to gently note that a sample size of 816 participants is comparable to contemporary studies in this field. For example:
>
> * Wang et al. (2025) used a sample of $N = 800$.
> * Petrov et al. (2024) used samples of $N = 150$ and $N = 1000$.
> * Park et al. (2024) used a sample of $N = 1052$.
> * Aher et al. (2023) used samples of $N < 100$ per experiment.
> * Choi & Li (2024) used a sample of $N = 99$.

---

> ### Author Response · Authors · 2025-11-21
> **Response to Reviewer GtxT (Part 2/2)**
>
> **2. On the Influence of Alignment and the Use of Base Models:**
>
> We thank the reviewer for suggesting that the "correlation amplification" phenomenon could be a result of the alignment phase (e.g., RLHF) where models are trained to simulate an agreeable personality.
>
> During our experimental design, we once considered using pre-trained base models but realized that they were not suitable for this study. We have two considerations:
>
> * **Task Incompatibility:** The pre-training task focuses on next-word prediction, whereas our experiment requires the model to possess a degree of instruction-following capability to understand the role-playing scenario and the response format. The pre-training objective does not align with our experimental goals.
> * **Reliability of Pre-trained Models:** It is well-documented that pre-trained models can suffer from significant issues such as hallucination, a lack of truthfulness, and the generation of toxic text (Ouyang et al., 2022; Lin et al., 2022; Ji et al., 2023). Conducting experiments on these models would make it difficult to draw reliable conclusions about psychological reasoning.
>
> Therefore, in line with established methods, we conducted our experiments on instruction-tuned models to ensure the results were both reliable and valid for our research question.
>
> The alignment phase, where models are trained to simulate an agreeable personality, may indeed introduce biases in models' responses, such as the "bias amplification" phenomenon (Taori & Hashimoto, 2023; Wang et al., 2024). However, being trained to be agreeable is unlikely to cause the "structural amplification" observed in our study, which reflects knowledge of the correlational structure of personalities.
>
> **3. On the Suggested References:**
>
> We are very grateful to the reviewer for providing these valuable references. We have cited them in the revised manuscript. In particular, we found the work by Sorokovikova et al. (2024) to be an excellent point of comparison for our findings.
>
> ---
> Below are the references we have mentioned:
>
> [1]Wang, Y., Zhao, J., Ones, D. S., He, L., & Xu, X. (2025). Evaluating the ability of large language models to emulate personality. Scientific reports, 15(1), 519.
>
> [2]Petrov, N. B., Serapio-García, G., & Rentfrow, J. (2024). Limited ability of llms to simulate human psychological behaviours: a psychometric analysis. arXiv preprint arXiv:2405.07248.
>
> [3]Park, J. S., Zou, C. Q., Shaw, A., Hill, B. M., Cai, C., Morris, M. R., ... & Bernstein, M. S. (2024). Generative agent simulations of 1,000 people. arXiv preprint arXiv:2411.10109.
>
> [4]Aher, G. V., Arriaga, R. I., & Kalai, A. T. (2023, July). Using large language models to simulate multiple humans and replicate human subject studies. In International conference on machine learning (pp. 337-371). PMLR.
>
> [5]Choi, H. K., & Li, Y. (2024). Picle: Eliciting diverse behaviors from large language models with persona in-context learning. arXiv preprint arXiv:2405.02501.
>
> [6]Ouyang, L., Wu, J., Jiang, X., Almeida, D., Wainwright, C., Mishkin, P., ... & Lowe, R. (2022). Training language models to follow instructions with human feedback. Advances in neural information processing systems, 35, 27730-27744.
>
> [7]Lin, S., Hilton, J., & Evans, O. (2022, May). Truthfulqa: Measuring how models mimic human falsehoods. In Proceedings of the 60th annual meeting of the association for computational linguistics (volume 1: long papers) (pp. 3214-3252).
>
> [8]Ji, Z., Lee, N., Frieske, R., Yu, T., Su, D., Xu, Y., ... & Fung, P. (2023). Survey of hallucination in natural language generation. ACM computing surveys, 55(12), 1-38.
>
> [9]Taori, R., & Hashimoto, T. (2023, July). Data feedback loops: Model-driven amplification of dataset biases. In International Conference on Machine Learning (pp. 33883-33920). PMLR.
>
> [10]Wang, Z., Wu, Z., Zhang, J., Guan, X., Jain, N., Lu, S., ... & Koshiyama, A. (2024). Bias Amplification: Large Language Models as Increasingly Biased Media. arXiv preprint arXiv:2410.15234.
>
>
> ---
>
> Once again, we thank you for your time and the constructive comments, which have significantly helped us strengthen our paper.

---

### Author Response · Authors · 2025-12-01
**Summary of Reviewer Concerns and Rebuttal Actions**

To the Area Chair,

We thank the reviewers (GtxT, Pih4, UCxr, GETG) for their highly constructive feedback. We are encouraged by the positive reception and the ​consensus on the interest and significance of our work​. Specifically, reviewers highlighted the novelty of "reframing" personality analysis from first-order prediction to second-order structural correlation (Reviewers Pih4, GETG) and praised the rigor, coherence, and depth of the experimental methodology (Reviewers Pih4, UCxr, GETG).

Below is a summary of the primary concerns raised and the specific actions we have taken to address them during the rebuttal period.

### 1. Reasoning vs. Semantic Pattern Matching

**Concern:** Reviewers Pih4 and UCxr asked for clearer evidence that models are performing genuine abstraction rather than simple semantic matching or exploiting pre-trained knowledge of scale names.

**Response & Action:**

* **Semantic Baseline Benchmark:** We added a comparative analysis against a Semantic Similarity Baseline. As shown in Figure 4, the LLMs' information selection aligns significantly closer with human-like conceptual factors than with surface-level semantics, demonstrating that their performance transcends simple pattern matching.
* **Prompting Clarification:** We explicitly clarified that predictive models were neither shown target scale names (preventing direct knowledge retrieval) nor explicitly prompted for Chain-of-Thought. This confirms that the observed reasoning was a spontaneous, emergent process acting solely on raw item content, ruling out methodological artifacts.

### 2.  Experimental Robustness & Methodology

**Concern:** Reviewers GtxT, UCxr and GETG raised questions regarding generalizability, prompt sensitivity, and the choice of instruction-tuned vs. base models.

**Response & Action:**

* **Robustness Analysis:** We highlighted our analysis showing that results remain stable across three distinct conditions: Standard Order, Random Order (shuffled inputs), and Single Question format. This stability effectively rules out both specific prompt artifacts and potential data contamination.
* **Generalizability:** We expanded our reporting to include scatter plots for all 12 evaluated LLMs (Appendix B.2), demonstrating that structural amplification is a consistent property across different models.
* **Model Justification:** We clarified that instruction-tuned models were necessary because base models lack the instruction-following capability required for the strict role-play and formatting constraints of the psychometric tasks.

### 3. Validity of "Structural Amplification" (Reliability Artifacts)

**Concern:** Reviewer Pih4 insightfully suggested investigating the role of measurement reliability in the observed amplification effect, noting that the disparity between clean model signals and noisy human data could be a driving factor.

**Response & Action:**

* **Reliability & Attenuation Correction:** We confirmed that LLMs exhibit significantly higher internal consistency (Cronbach’s Alpha) than humans. Crucially, performing a correction for attenuation on human data significantly reduced the amplification coefficient ($k$). This analysis strengthens our "Idealization Hypothesis", providing robust empirical evidence that the amplification stems from the models systematically filtering out statistical noise to simulate a "true score" psychological structure.


### 4. Expanded Related Work & Contextualization

**Concern:** Multiple reviewers (GtxT, Pih4, GETG) requested a more comprehensive review of prior literature to better situate our contribution.

**Response & Action:**

* **Literature Update & Novelty Clarification:** We have significantly expanded the Introduction and Related Work sections to incorporate all suggested references. By explicitly contrasting our work with these studies, we have strengthened the argument for the novelty of our "second-order" structural analysis, which was recognized by reviewers as a novel and interesting reframing. Specific comparative discussions regarding these works have been integrated into the manuscript.

### 5. Completeness and Reproducibility

**Concern:** Reviewers noted a need for transparent data sharing to ensure the reproducibility of our analyses.

**Response & Action:**
We have uploaded a complete reproducibility package to our repository, which now includes:

* The full anonymized human dataset ($N=816$) and all psychological questionnaires used.
* Comprehensive Model Artifacts: To fully document the reasoning process, we included all raw LLM outputs, reasoning traces, extracted summaries, mapping results from the annotation models, and the specific model outputs for the three information conditions (Section 4.2).

We believe these revisions have substantially strengthened the paper and addressed the core technical and methodological questions. We hope this summary assists in your final decision.

Sincerely,

The Authors

---

### Meta-Review · Area_Chair_yy4c · 2026-01-08

**Summary:**

This paper investigates the capability of Large Language Models (LLMs) to infer the latent correlational structure of human psychological traits. The authors propose a "second-order" evaluation paradigm: rather than predicting specific item responses for a user (first-order), the model is given a user's Big Five personality profile and asked to role-play that user across 9 other psychological scales. The reviewers were generally positive about the novelty of the ``second-order'' framing, but raised significant technical concerns regarding the validity of the ``amplification'' effect and the underlying mechanisms.

The reviewers are largely in consensus regarding the interest and significance of the work (Scores 8, 8, 6, 4), and the most critical reviewer's concerns (Pih4) regarding reliability artifacts were directly and effectively addressed by the new analysis. The paper is well-executed and provides a strong foundation for future work in synthetic user simulation. Therefore, I recommend accepting this paper.

**Reviewer Concerns:**

(1) (Reviewers Pih4, UCxr) Reviewers questioned whether the model's performance was merely due to surface-level semantic similarity between questionnaire items or knowledge of scale names, rather than genuine psychological reasoning. The authors addressed this by introducing a Semantic Similarity Baseline, demonstrating that LLM conceptual selection aligns significantly closer to human factors than semantic embeddings do. They also clarified that target scale names were hidden, preventing direct knowledge retrieval, which is satisfactory to me.

(2) (Reviewer Pih4) A critical concern was that the ``amplification'' (slope > 1) might simply be an artifact of comparing noise-free model predictions against noisy human data (attenuation), rather than a meaningful ``idealization.'' The authors performed a Correction for Attenuation on the human data. This analysis showed that when human measurement error is accounted for, the amplification coefficient drops significantly toward 1.

(3) (Reviewers GtxT, GETG) Reviewers requested checks on prompt sensitivity, model selection (Base vs. Instruct), and broader related work. The authors demonstrated robustness across Random Order and Single Question setups, ruling out context artifacts.

**Reviewer Scores:**

The reviewers are largely in consensus regarding the interest and significance of the work (Scores 8, 8, 6, 4), and the most critical reviewer's concerns (Pih4) regarding reliability artifacts were directly and effectively addressed by the new analysis.

So the reviewer Pih4 is expected to increase his score to 6. The final overall score could be 8, 8, 6, 6  if reviewers had been able to participate fully in the discussion.

---

### Decision · Program_Chairs · 2026-01-26

Accept (Poster)